# Wastewater surveillance of SARS-CoV-2 from aircraft to citywide monitoring

Mariel Perez-Zabaleta[1,5], Carlo Berg [2], Neus Latorre-Margalef[3], Isaac Owusu-Agyeman[1], Ayda Kiyar[1], Helene Botnen [3], Caroline Schönning[3], Luisa W. Hugerth [4] ✉ & Zeynep Cetecioglu [1] ✉

Wastewater monitoring is highly efficient in SARS-CoV-2 surveillance for tracking virus spread through travel, surpassing traditional airport passenger testing. This study explored the links between SARS-CoV-2 contents and variants from aircraft to city, assessing the impact of detected variants from international travellers versus the local population. A total of 969 variants using next-generation sequencing (NGS) were examined to understand the links between−aircraft, Arlanda airport, wastewater treatment plants (WWTPs), and Stockholm city−and compared these to variants detected in Stockholm hospitals from January to May 2023. SARS-CoV-2 contents in WWTPs reflected local infection rates, requiring analysis from multiple plants for an accurate city-wide infection assessment. Variants initially detected in aircraft arriving from China did not spread widely during the study period. RT-qPCR is adequate for the detection of specific variants in wastewater, including Variants Under Monitoring. However, NGS remains a powerful method for identifying novel variants. Wastewater monitoring was more effective than clinical testing in the early detection of specific variants, with notable delays observed in clinical surveillance. Furthermore, a broad range of variants are detected in wastewater that surpasses clinical tests. This underscores the vital role of wastewater-based epidemiology in managing future outbreaks and enhancing global health security.

SARS-CoV-2, identified in Wuhan in December 2019, was declared a global pandemic by March 2020[1]. The virus evolves rapidly, with variants classified by the World Health Organization (WHO) as variants under monitoring (VUMs), variants of concern (VOCs) and variants of interest (VOIs)[2]. Omicron, more transmissible than previous variants, dominates the current landscape, accounting for more than 98% of the available sequences since February 2022[2,3]. Since March 2023, Omicron sub-lineages are considered independently as VUMs, VOIs or VOCs[2].

Mass testing was initially implemented to manage SARS-CoV-2. However, with the advent of the Omicron variant and increased immunity[4], testing declined, impacting data accuracy. Many countries have since focused on testing high-risk groups and hospital patients. Some have introduced airport testing for travellers[5]. Given the high cost of PCR testing[4], wastewater-based epidemiology (WBE) has been adopted as a cost-effective alternative. As of January 2024, WBE programmes were active in at least 72 countries[6]. WBE can detect SARS-CoV-2 variants earlier than clinical surveillance, as individuals typically

[1]Department of Industrial Biotechnology, School of Engineering Sciences in Chemistry Biotechnology and Health, KTH Royal Institute of Technology, AlbaNova University Center, Stockholm, Sweden. [2]Department of Microbiology, Public Health Agency of Sweden, Solna, Sweden. [3]Department of Communicable Disease Control and Health Protection, Public Health Agency of Sweden, Solna, Sweden. [4]Department of Medical Biochemistry and Microbiology, Science for Life Laboratory, Uppsala University, Uppsala, Sweden. [5]Present address: Research Group for Food Microbiology and Hygiene, National Food Institute, Technical University of Denmark, Kongens, Lyngby, Denmark. ✉e-mail: luisa.hugerth@scilifelab.se; zeynepcg@kth.se

seek medical attention one to 2 weeks after infection[7–9]. In addition, surveillance of the wastewater from aircraft and cruise ship can provide information about the importation of SARS-CoV-2[10,11].

SARS-CoV-2 variants have spread globally through international travel[12], with initial Omicron cases identified at international airports[13,14]. Many countries reacted by imposing travel restrictions[15]. As conditions improved, these restrictions were gradually lifted, some as early as 2021 and others in 2022. China, with stringent COVID-19 regulations, reopened its border in January 2023[16]. However, the rise in air traffic and the emergence of new SARS-CoV-2 subvariants, such as the XBB lineage, have raised concerns about potential new COVID-19 outbreaks. The proposal to sample wastewater from aircraft lavatories and airport sewage systems has been put forward as a response. This approach eliminates the need for traveller participation, reducing ethical concerns[17]. Early identification and tracking of emerging variants at entry points could allow prompt public health actions, potentially mitigating variant spread. Previous studies on SARS-CoV-2 surveillance on aircraft demonstrated the feasibility of detecting SARS-CoV-2 concentrations using qPCR[10,11,18–20]. However, only a few of these studies monitored SARS-CoV-2 variants[21–23] and none examined more than 44 variants.

Sweden's government tasked the Public Health Agency of Sweden (PHAS) with risk assessment and monitoring of incoming variants when China lifted its COVID-19 restrictions[24]. Given the uncertain epidemiological situation in China by the end of 2022, European governments adopted a cautious approach, requiring proof of negative tests for travellers from China from January 7th to February 19th, 2023[25]. The EU Commission and European Centre for Disease Prevention and Control issued recommendations for testing travellers and wastewater from aircraft lavatories[26]. In response to the given recommendation, this study was promptly initiated. Wastewater sampling from aircraft tanks originating from China started on January 13, 2023, at Stockholm Arlanda. Notably, the first aircraft to arrive at Arlanda from China after the EU recommendation was immediately subjected to sampling. Arlanda is Sweden's largest airport, serving nearly 27 million passengers annually before the COVID-19 pandemic. This number obviously declined during the pandemic but there was a recovery in passenger traffic in 2022, with a recorded volume of 22 million[27]. This study evaluated SARS-CoV-2 contents and variants, determining lineages in wastewater from aircraft lavatories and Arlanda's airport sewage facilities and compared these with local population data to assess what information could be gained at each of these levels. While other studies focused solely on monitoring SARS-CoV-2 concentration on aircraft[10,11,18], detecting only a few variants[21–23] and covering small areas such as aircraft and airport[18,21], the current study adopted a singular approach by linking a large study area (aircraft, airport, WWTP and city) and monitoring SARS-CoV-2 concentration and detecting numerous variants. This comprehensive approach provides a valuable understanding of almost 1000 virus variants detected from small to big catchment areas, underscoring the high potential of WBE in disease control.

## Results

### Normalised SARS-CoV-2 quantities at different sampling sites

The qPCR negative controls (tap water and NTC) showed no amplification, confirming the absence of contamination. The positive controls (SARS-CoV-2 IDT, PMMoV IDT and cross-plate) produced the expected amplification results, verifying the accuracy and efficiency of the qPCR procedure. These results demonstrate the reliability of the qPCR process. PMMoV concentrations exhibited stability across all samples, indicating a uniform distribution of population loads and effective RNA extraction processes. The average PMMoV Cq value across all inlets of the WWTPs in Stockholm is $23.58 \pm 0.99$ and a variance of 0.98 ($1.1 \times 10^7 \pm 6.4 \times 10^6$ gene copies/L and variance $4.1 \times 10^{10}$ gene copies/L). However, aircraft samples showed increased PMMoV variability due to the fluctuating faeces-to-water ratio as vacuum toilets have minimal water usage, with an average PMMoV Cq value of $23.65 \pm 3.07$ and a variance of 9.43 ($2.4 \times 10^7 \pm 5.8 \times 10^7$ gene copies/L and variance $3.3 \times 10^{15}$ gene copies/L).

Käppala data was statistically evaluated against aircraft and airport data (Fig. 1c) and Stockholm region (Fig. 2a) to determine the correlations. The N-gen/PMMoV ratio in the aircraft and airport did not follow a clear trend and were dependent on the sampling occasion (Fig. 1a, b). The highest detected content in aircraft was in weeks 8, 16 and 17. Despite using 80 mL of wastewater instead of 40 mL to increase viral concentrations, N-gene/PMMoV ratios in aircraft remained near the detection limit during weeks 4, 7, 9 and 18. The airport had its highest N-gen/PMMoV ratio in week 17, which concurs with the highest peak of Käppala (Fig. 1). Based on the Shapiro-Wilk test, the data for Stockholm and Käppala follow a normal distribution. Statistical analyses showed a moderate positive Pearson correlation (0.60) between the viral contents at the airport and Käppala. However, no significant correlation was found between Käppala and Stockholm city ($r = 0.37$, $P = 0.095$). This aligns with the fact that Käppala serves only a third of Stockholm's population, while Henriksdal, the largest WWTP in Stockholm, serves 41% and has the highest infection rate (Fig. 2b).

Additionally, well-known primers for SARS-CoV-2 detection (N1, N2 and N3) were compared to study whether any of them could have a better detection limit for these samples. The results demonstrated that the N2 and N3 primers can cover more SARS-CoV-2-related viruses than the N1 primer when the sample has high viral contents. However, neither used primers showed enhanced detection capabilities in samples with low viral contents such as aircraft samples (Fig. S3, supplementary material). The N3 primers were preferred over N2 due to evidence from another study indicating that N3 exhibited superior performance and better detection limits with stool samples[28].

Furthermore, the detection of SARS-CoV-2 in aircraft samples was evaluated using both the TaqMan and SYBRGreen methods. The

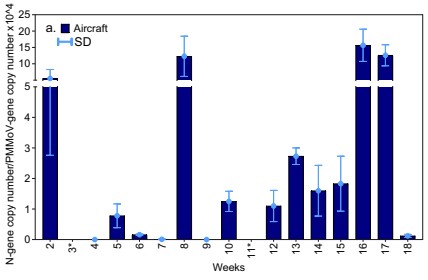
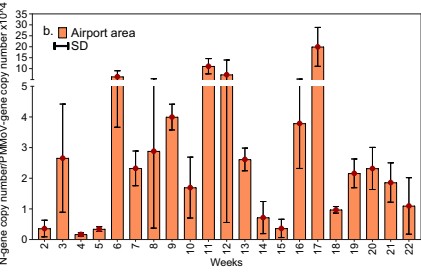
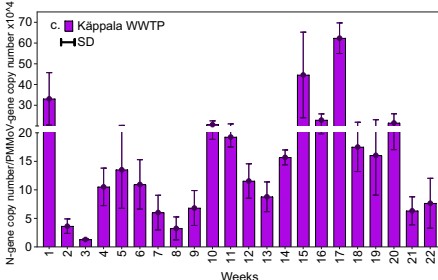

**Fig. 1 | SARS-CoV-2 content presented as N-gene copy number per PMMoV gene copy. a** Aircraft **b** Måby station which corresponds to the Airport area **c** Käppala WWTP. (*) No aircraft sample in weeks 3 and 11. Two biological replicates and two technical replicates were analysed for each data point. Data are presented as mean values ± standard deviation (SD).

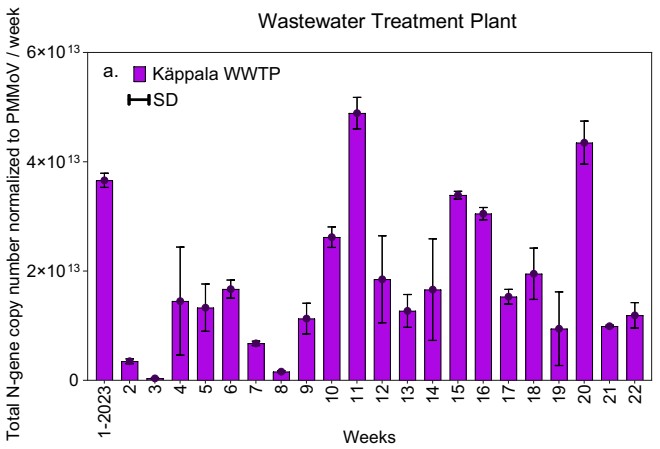
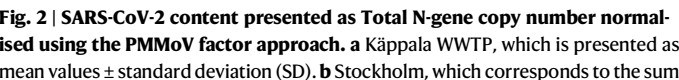
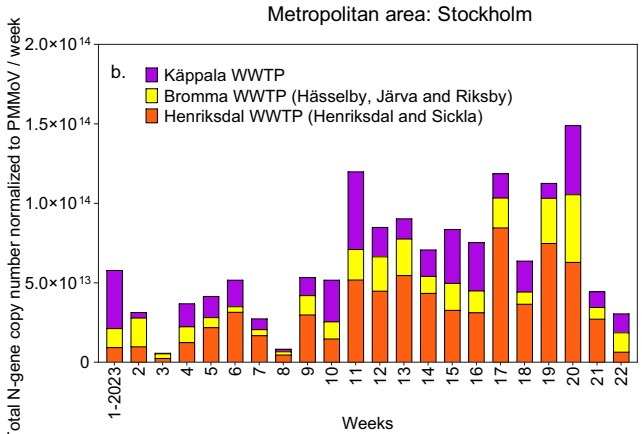

**Fig. 2 | SARS-CoV-2 content presented as Total N-gene copy number normalised using the PMMoV factor approach. a** Käppala WWTP, which is presented as mean values ± standard deviation (SD). **b** Stockholm, which corresponds to the sum of SARS-CoV-2 contents in the three main WWTPs: Käppala. Henriksdal and Bromma. Two biological replicates and two technical replicates were analysed for each data point.

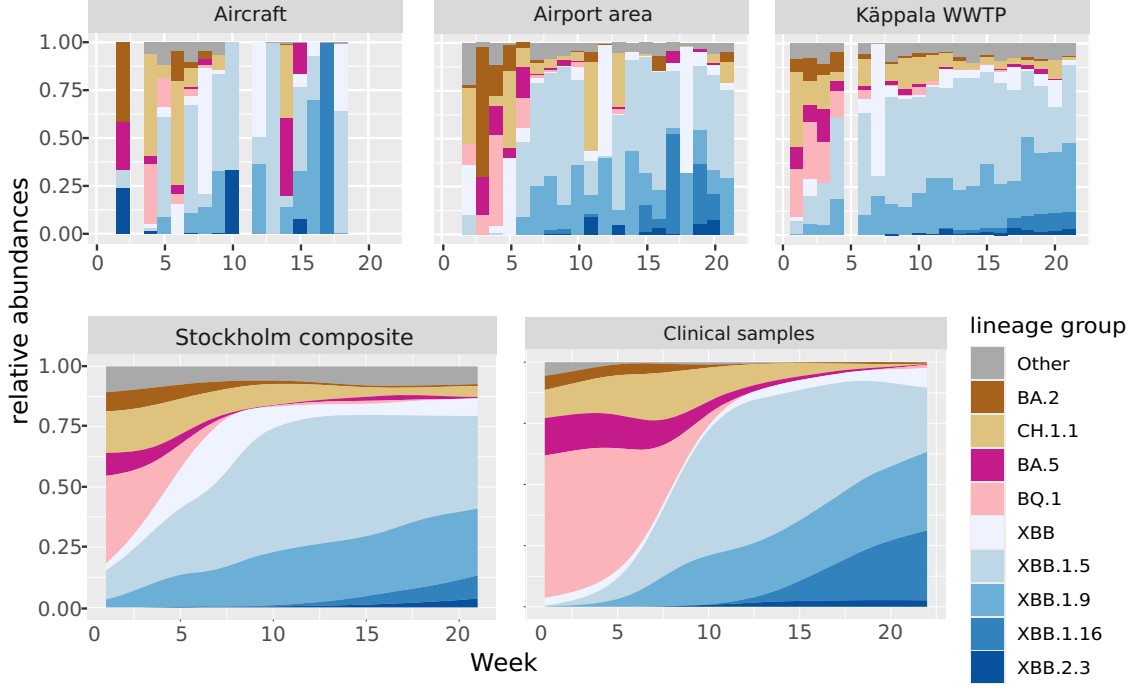

**Fig. 3 | SARS-CoV-2 lineage groups detected at different catchment areas including the incoming aircraft from China, the airport area, its associated WWTP: Käppala, a composite sample of all Stockholm WWTP and healthcare data.** The data for patients and composite samples are smoothed; since this approach would not work in the aircraft data, it is not done for the other smaller regions either.

comparison revealed no clear differences between the two methods (Fig. S4). At times, both methods detected similar amounts of SARS-CoV-2, while in other instances, TaqMan reported lower or higher levels than SYBRGreen. The SYBRGreen was preferred over TaqMan because with suitable primer sets such as N3, SYBRGreen has been shown to perform comparably or even better than the TaqMan method in terms of sensitivity, which is crucial for detecting low levels of viral RNA in wastewater samples[29].

### Sequencing data at different sampling sites
The patient data were used to explore which variants/lineages circulated in the population at the same time as active sampling at Arlanda airport took place. At a high aggregation level, the lineage profile

detected in the airport, Käppala WWTP and Stockholm composite sample were remarkably similar (Fig. 3). The clinical data had a delay, which could be due to the time elapsed between the infection occurring and the subsequent sampling upon hospitalisation. This was most notable in the persistent detection of BQ.1 and BA.5 variants in patients, which was observed for 5 weeks longer than in wastewater. The aircraft data seemed much more stochastic, which is expected when the corresponding population (up to 312 people) was different in each flight and more than 1000-fold smaller than in e.g. Käppala. Additionally, the data from the aircrafts, having lower coverage and depth, was more often not fully assigned by Freyja, receiving a truncated lineage. This affected 35% of the total sequence variants in the aircraft data, in comparison to 4.3% on average for the other sites.

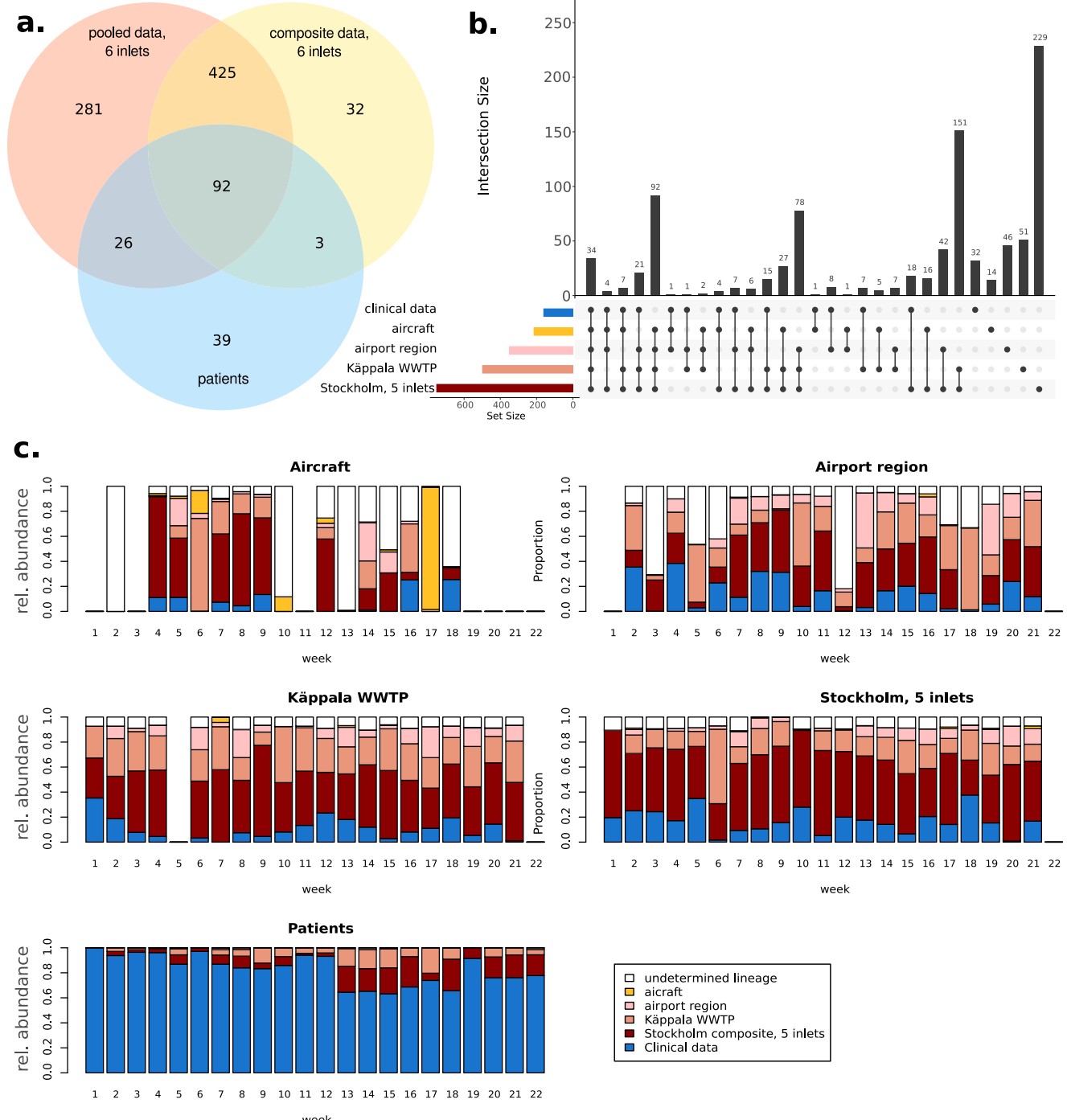

**Fig. 4 | Overlap between SARS-CoV2 lineages detected in different sites. a** Venn diagram showing the overlap between lineages detected in patients and those detected in Stockholm wastewater, considering either a composite sample (mixed in the lab) or pooled sequencing results from each inlet. **b** Upset plot of the five analysis levels, from aircraft to patients. An Upset plot shows a horizontal bar plot of the total number of lineages found at each site and on the vertical bar plot the number of lineages found at each combination of sites. Combinations are depicted as coloured dots corresponding to the sites on the left. **c** Bar plots showing the relative abundance of lineages, coloured after their first site of detection within our dataset. Lineages that could not be fully determined are left blank since their origin is uncertain.

There was a very high correlation between Käppala and the other inlets of the WWTPs in Stockholm (all symmetric Procrustes correlations >0.95, all $p < 0.001$) (Fig. S7).

To study whether lineages may have been introduced to Stockholm through the airport region, it is not enough to assess these broad levels, since many viral strains belonging to the same clade could have been independently introduced to the study area by different people. Instead, we assessed it at the deepest lineage level available. Since the composite sample did not capture the full diversity of lineages from the WWTP (probably as a result of sequencing depth; Fig. 4a), downstream comparisons were performed using an in silico pool of 5 inlets; the Käppala inlet, being downstream of the airport region, is presented separately. Most wastewater lineages were detected in all WWTP samples ($n = 197$), but

all WWTP samples also presented exclusive lineages as can be observed in Fig. S8 ($n = 28$–$38$ and $n = 73$ in Käppala).

The number of lineages captured at each catchment area is in direct proportion to the population included (Fig. 4b). Each aggregation level revealed unique lineages. While most lineages found in patients were present in all sites ($n = 34$) or all except the aircraft ($n = 21$), another 32 lineages were only observed in patients during this period. Ten lineages were detected in patients and the aircraft or airport region, but not in the other wastewater samples.

When considering the relative abundance of lineages at each site, clinical samples were dominated by lineages found first in patients, although lineages first detected in wastewater also make up about half of the total lineages from week 13 and onward (Fig. 4c). Conversely, the Stockholm pooled sample was consistently dominated by lineages which were first detected in wastewater. Lineages first detected in the aircraft and airport region can be found at all other sites but never became numerically dominant.

## Discussion

This research undertook a detailed examination of the presence and diversity of SARS-CoV-2 in wastewater from various sources, including aircraft, airport region, wastewater treatment plants and Stockholm City. Almost 1000 SARS-CoV-2 genomic variants were identified in this study, aiming to decipher the complex interconnections between different catchment areas. Aircraft wastewater monitoring is a promising tool for SARS-CoV-2 surveillance, matching the effectiveness of randomly testing 20% of passengers and outperforming respiratory swab tests[19]. While several studies have focused on this topic, the influence of SARS-CoV-2 contents and variants of aircraft and airport regions on urban areas remains unexplored. This study could also serve as a valuable reference for further research in other transportation networks. Here, aircraft- and airport-detected variants were contrasted to those in the population. Additionally, a thorough examination was undertaken to compare the variants identified in Stockholm's wastewater with those documented in the clinical surveillance.

Our results on SARS-CoV-2 concentrations showed that wastewater surveillance from the WWTP (Käppala, which covers 33% of the population) only represents infections in that area and cannot be extrapolated to Stockholm city. For a complete picture of Stockholm's infection, it is necessary to consider Bromma and Henriksdal WWTPs contents as well. However, when the SARS-CoV-2 variants were examined, the lineage profile detected in Käppala was similar to that of Stockholm. The number of WWTPs required for effective city-wide infection assessment is highly dependent on local conditions. Key factors include the size of the sewer-shed, which determines population coverage and the type of sewer system, whether separated or combined. In some instances, a single large WWTP that covers 70-80% of the population in a large city may be sufficient for comprehensive monitoring.

Monitoring wastewater has proven to be an effective method for detecting certain variants compared to clinical data, which showed a noticeable delay[30,31]. A relevant finding of the study was that wastewater samples exhibit six times more variant diversity than clinical cases, and remarkably, wastewater monitoring can detect these variants up to 5 weeks before their appearance in clinical cases. Most of these variants are likely shed by asymptomatic carriers or those with mild symptoms, but this still proves that a VOC with international spreading would likely be detected earlier in wastewater than in the clinics. Furthermore, the study found a direct correlation (Pearson $r = 0.98$) between the population size of a specific sample and the number of detected variants. In other words, wastewater samples from larger populations were associated with a higher number of detected variants. This trend was observed across all areas, with Stockholm showing the highest number of variants, followed by WWTPs, airport

regions and finally, aircraft. Not only does clinical data report fewer variants than wastewater, but wastewater surveillance also has the advantage of early detection of these variants in this study. For instance, variants XBB.1.5 and XBB.1.9 were detected in wastewater as early as the 1st week of 2023. In contrast, these variants were only considerably detected in clinical cases from the 5th week onwards, highlighting the pivotal role of wastewater surveillance in the early detection of SARS-CoV-2 variants and emphasising its importance in our public health strategies. Given the cost-effectiveness of WBE, its value lies in complementing clinical sampling efforts[32], especially in situations when clinical samples are scarce or difficult to obtain such as from travellers. This also suggests that, for monitoring, a balance must be achieved between coverage and granularity of the wastewater sources being analysed. Single-person and single-aircraft approaches do not scale well, while we also found that combining all samples before analysis was not as sensitive as analysing each inlet separately. WBE provides also cost-effective baseline data to follow the epidemiology and transitions in variant diversity with the possibility to scale up clinical testing upon certain events such as the global recognition of a new VOC or VUM. In a public health context, the spread of a new VOC or VUM and its expected arrival in a country or region can be monitored more proactively, as shown previously[33].

There are several challenges in using wastewater for SARS-CoV-2 surveillance such as detection limit, long-term shedding and the influence of chemicals and detergents added in the toilet tanks on virus particle recovery[22]. Prolonged shedding of viral RNA post-infection could indeed contribute to positive detections, but it could also hinder efforts to obtain robust sequences[23]. In the case of aircraft concentrations observed in this study, being near the assay's limit of detection (LOD) (weeks 4, 7, 9 and 18) suggests that measures taken by the EU may have effectively restricted the movement of travellers who tested positive. Aircraft samples in this study were grabbed but are thought to be representative of the whole flight because the wastewater was mixed during transport from the aircraft to the lavatory service truck[23]. However, even in long-haul flights such as the ones analysed here, not all travellers defecate during the flight, which could influence the representativeness of the sample[20,23]. Therefore, broadening the scope of the sampling location could improve sensitivity and decrease the time to first detection, as defined by St-Onge et al.[34]. Additionally, there are problems intrinsic to metagenomic sequencing, including fragmented data, which makes the discovery of recombinant lineages fraught; and uneven sequencing depth, which could mask differences between lineages, if their defining SNV are not well covered.

Our study showcases the use of WBE as a tool for evaluating viral diversity across various locations and timeframes. By examining multiple levels, from aircraft to regional scales simultaneously, WBE offers a unique approach to detect trends and anticipate the introduction of known variants, complementing the clinical testing. Examining 969 variants in wastewater samples and aligning them with clinical data sets, our study highlights the importance of wastewater sampling in transportation hubs and various environments. To manage the risk of new variant introductions via global transport networks, establishing surveillance measures at critical entry points such as harbours or train stations could be highly valuable.

This study underlines the broad scalability and flexibility of wastewater surveillance. Especially in the light of limited clinical testing, wastewater surveillance demands comparatively fewer resources at both smaller and larger scales yet providing comprehensive information on the variant diversity. This pilot project has shown that wastewater monitoring can detect variants up to 5 weeks ahead of clinical data, identifying six times more variants than clinical surveillance alone, showing valuable results on variant composition and providing crucial information about the import of variants/pathogens due to travel activities. Our investigation suggests that wastewater sampling

at airports and a complementary surveillance system could be a viable approach for monitoring not only SARS-CoV-2 but potentially other pathogens as well. In contrast to an individual clinical patient sample, one wastewater sample can potentially be used for the monitoring of multiple different pathogens at the same time, further enhancing scalability and cost-effectiveness. While setting up a surveillance system presents challenges, including legal implications and practical aspects of implementation and maintenance, the potential public health benefits make it a compelling pursuit and could play a pivotal role in disease surveillance and control, significantly bolstering global health security.

## Methods

### Wastewater sampling

From January to May 2023, weekly wastewater samples were collected from aircraft, Arlanda Airport Terminal 5, Måby station (covering the airport region), Käppala WWTP, Bromma WWTP and Henriksdal WWTP (Figs. S1 and S2, supplementary material). Stockholm has an urban population of around 1.6 million and a metropolitan population of ~2.4 million. Samples were collected from three main municipal wastewater treatment plants in Stockholm: Bromma WWTP (serving around 377,500 residents, 18% of the population), Henriksdal WWTP (serving about 862,100 residents, 41% of the population) and Käppala WWTP (serving around 700,000 residents, 33% of the population). Bromma WWTP has three inlets: Hässelby, Riksby and Järva, while Henriksdal WWTP has two inlets: Sickla and Henriksdal. All airport wastewater goes to Käppala (Fig. S1) and this WWTP has only one inlet. The samples from the six inlets and the airport were flow-compensated and ~500 mL of each sample was collected over 24 h from Monday to Tuesday every week, using stationary flow-proportional samplers SP5 B (MAXX Mess- und Probenahmetechnik GmbH, Germany) in Henriksdal WWTP, TP5 W (MAXX Mess- und Probenahmetechnik GmbH, Germany) in Bromma WWTP and Efconomy (Efcon Water B.V., The Netherlands) in Käppala WWTP.

Stockholm's samples were collected from week 1 to 22 of 2023 while aircraft samples from week 2 to 18, except for the 3rd and 11th weeks when flights were cancelled. A direct flight from China carrying 312 passengers and with a duration of ~9 h was weekly monitored, on Fridays from week 2 to 9 and on Tuesdays from week 10 to 18. Aircraft wastewater samples were collected after thorough mixing during transport from the aircraft to the lavatory service truck. Samples were then taken from the top of the lavatory service truck immediately after the transfer. Grab samples, reflecting the microbiological status at the time of collection, were taken from Stockholm's Arlanda Terminal 5 with data available in supplementary material (Figs. S4 and S5). All samples were transported to the lab on cooling boxes and with ice packages. Samples from aircraft and airport (Måby and Arlanda terminal 5) were processed for concentration and RNA extraction in the same day (less than 24 h after sampling), while samples from the six inlets of the WWTP were processed within 24 h after sampling.

### Wastewater concentration and RNA extraction

Wastewater samples were concentrated using a Maxwell RSC Enviro TNA Promega Kit following the manufacturer's instructions, with some exceptions. From the 500 ml of wastewater sample collected weekly, 40 ml aliquots were used for analysis and treated with a protease solution and centrifuged to remove precipitated proteins and solids present in the sample. Then, the supernatant was filtrated and eluted to 500 μL using a column-based system and then loaded into a cartridge provided by the kit. Afterwards, total nucleic acid (TNA) was extracted using Maxwell RSC Instrument (Promega Biotech AB, Sweden) and Maxwell RSC Pure Food GMO programme was selected. The elution volume was 80 μL using nuclease-free water. Two independent biological replicates were concentrated and RNA extracted per sample. Two tap water samples were used as negative controls in each

extraction set (extraction of 16 samples). To ensure unbiased analysis, samples were labelled with specific codes for each location and week, and these codes were randomised during the wastewater concentration and TNA extraction processes.

Due to the relatively low SARS-CoV-2 content in aircraft samples, a spatial composite of aircraft wastewater was implemented from week 5 onwards to enhance the likelihood of positive detection and increasing the amount of virus before RT-qPCR analysis. Four independent wastewater samples, each with a volume of 40 mL, were concentrated and eluted to a final volume of 500 μL using the Promega kit. The four resulting concentrates of 500 μL each were then divided into two groups, and the samples within each group were combined, yielding ~1000 μL per group. These combined samples were subsequently used for RNA extraction, producing two independent biological replicates, each with a volume of 80 μL.

### Weekly monitoring of SARS-CoV-2

SARS-CoV-2 contents were determined via reverse transcriptase quantitative polymerase chain reaction (RT-qPCR)[35,36]. The reaction was performed using SYBR Green one-step kit (Bio-Rad) according to the manufacturer's instructions, with the modification of adding 2 μL of 4 mg/ml Bovine Serum Albumin (BSA) (Thermo Scientific™) to reduce PCR inhibitors and enhance efficacy, for a final reaction volume of 20 μL. For SARS-CoV-2 detection, N3 primers targeting the nucleocapsid (N) protein were used, FW: 5'-GGGAGCCTTGAATA-CACCAAAA-3' and RV:5'-TGTAGCACGATTGCAGCATTG-3'[37]. Pepper mild mottle virus (PMMoV) was also quantified in the samples and used for data normalisation as previously described[36]. The primers used for PMMoV quantification were forward, 5'-GAGTGGTTTGACCTTAA CGTTTGA-3'; reverse, 5'-TTGTCGGTTGCAATGCAAGT-3'[38]. Nuclease-free water was included as no-template control (NTC) for all qPCR reactions. Two tap water samples were concentrated, and RNA was extracted and analysed alongside the weekly samples to assess potential contamination during handling. SARS-CoV-2 DNA (2019-nCoV_N_Positive Control, IDT, Cat. 10006625), and a constructed plasmid containing the appropriate target for PMMoV (IDT, custom MiniGene 25-500 bp) were used as positive controls and to create the standard curves. A control sample of RNA from wastewater (cross-plate controls) with known SARS-CoV-2 and PMMoV concentrations was used in each qPCR analysis for reproducibility and quality control. Cross-plate controls were stored at −80 °C in aliquots. A new batch was prepared when the quantification cycle (Cq) shift of 0.5–1 was detected, with each batch typically used for up to 8 weeks. The cross-plate control was employed to evaluate the long-term precision, which refers to the variation in results between runs, in accordance with MIQE guidelines[39]. The standard deviation of the cross-plate controls was calculated between runs. Results were accepted if the standard deviation of the Cq values was less than 0.5. Two independent technical replicates were measured for each biological replicate and each control sample (positive, negative and cross-plate controls) using qPCR. Results were accepted if the standard deviation of Cq values between technical replicates was less than 0.5. Thermal cycling (50 °C for 10 min, 95 °C for 30 s, followed by 45 cycles of 95 °C for 10 s and 60 °C for 30 s) and melting curve detection were performed (65 °C to 95 °C with an increment of 0.5 °C for 5 s) on CFX96 Touch System (Bio-Rad). Reactions were considered positive if the fluorescence crossed the established threshold before 40 cycles (if Ct was less than 40) and if a single melting peak was observed at the correct temperature. The threshold was set automatically using CFX Manager™ Software. The standard curves yielded calculated efficiencies of 98% and 90% for the N3 and PMMoV qPCR assays, respectively, with corresponding slopes of −3.38 and −3.59. The coefficient of determination ($R^2$) values were 0.999 and 0.998, respectively. The LOD and limit of quantification (LOQ) for the qPCR assays was determined using the standard deviation method, which involves the standard deviation of the

response (y-intercepts of the regression lines) and the slope of the calibration curve. The LOD and LOQ of the qPCR assays used for N3 were 0.4 copies/ml of wastewater sample and 5.3 copies/ml, respectively, and for PMMoV, they were 0.4 copies/ml and 4.1 copies/ml, respectively. Inhibition testing was conducted utilising the Cq dilution method, as recommended by the MIQE guidelines[39]. The results demonstrated high efficiencies and strong R² correlation coefficients for the standard curve. An example of the inhibition test is provided in the supplementary material (Fig. S9). A sample was considered positive if at least three out of four measurements were positive, including two technical replicates for each of the two biological replicates. If only two out of four measurements were positive, the samples were re-tested by qPCR. If the re-test yielded the same result and the melting curves were inconclusive, the samples were concentrated and tested again by qPCR.

SARS-CoV-2 concentrations in aircraft samples were near the detection limit for several weeks during the monitoring period. For this reason, we evaluated whether the detection limit could be improved by using TaqMan method or if there was any difference among the well-known primers targeting the N protein (N1, N2 and N3 primes). The methodology and results are presented in supplementary material.

## Calculations

The SARS-CoV-2 contents in this study are presented as either N-gene copy number adjusted for variations in PMMoV contents per week (PMMoV factor, Fig. 2) or N-gene copies per PMMoV gene copies ×10⁴ (N-gene/PMMoV ratio, Fig. 1). The PMMoV factor and N-gene/PMMoV ratio calculation methods were previously described by Perez-Zabaleta et al.[36]. The PMMoV factor calculation method, which adjusts for changes in flow rates, has been utilised to analyse data from Käppala and Stockholm. This approach combines data from various inlets to provide the total SARS-CoV-2 detected in the Stockholm region. The N-gene/PMMoV method was not used to plot Stockholm data due to its inability to account for variations in flow rates at each testing location, which could lead to inaccurate results. For locations such as aircraft, Arlanda Terminal 5 and Måby (airport), where flow rate data is unavailable, the N-gene/PMMoV ratio method was used to present the results. However, these results were kept separate and not combined with each other. For reference, Fig. S6 was included in the supplementary material to present the data of the five locations without normalisation (PMMoV or flow rate).

## DNA sequencing, variant calling and lineage determination

After the wastewater concentration and RNA extraction steps, 25 µL of purified RNA was shipped to the Uppsala Genome Center (Science for Life Laboratory, Dept. of Immunology, Genetics and Pathology, Uppsala University) for sequencing. The two biological replicates were mixed in equal amounts (12.5 µL) and sent for sequencing per sampling point. These sampling points included the six inlets of the WWTPs (Henriksdal, Sickla, Hässelby, Järva, Riksby and Käppala), aircraft, Arlanda Terminal 5 and Måby (airport region) samples. For Stockholm composite samples, 4.5 µL of each WWTP inlet were pooled, resulting in a volume of 27 µL of sample per week. For weeks 7, 8 and 9, the mixed sample from Stockholm was not sequenced, and data was pooled in silico. Additionally, in silico pools including all six inlets or five inlets excluding Käppala, were created when needed and specified in the results.

The samples were then reverse-transcribed with SuperScriptVILO cDNA synthesis (Thermo Fisher Scientific, Waltham, MA, USA) and the libraries were prepared using the AmpliSeq SARS-CoV-2 panel (Thermo Fisher Scientific) on two S5 540 chips. The libraries were barcoded and pooled into 32-plex and sequenced on the Ion S5XL system (Thermo Fisher Scientific). Processing raw sequencing reads, and single nucleotide variant (SNV) calling was done through the

Torrent Suit AmpliSeq SARS-CoV-2 pipeline according to the manufacturer's instructions. The resulting BAM files were used for post-processing with the Freyja pipeline, v.1.4.8[8]. Briefly, 'Freyja variants' were used to track SNVs against the Wuhan-2019-nCoV reference genome. Then, to convert the depth of each SNV into a likely relative abundance of PANGO lineages[40], 'Freyja demix' was used with barcode version 12_12_2023-00-48, a depth cutoff of 10x, a minimum relative abundance (eps) of 0.001 and only confirmed variants[41–43]. Sequences are deposited at ENA under https://www.ebi.ac.uk/ena/browser/view/PRJEB61810. For statistical analyses, only variants that could be fully assigned by Freyja, without the ambiguity markers 'Misc' or '-like' were considered. Figures 3, 4 and S7, S8 were created in R v. 4.3.1, with libraries VennDiagram 1.7.3 and UpSetR 1.4.0.

## Clinical SARS-CoV-2 variant data

Clinical infection data on COVID-19 patients were only available from Stockholm city since no epidemiological data corresponding to the aircraft, airport and Käppala were available. Test recommendations changed in February 2023, from including patients and healthcare personnel to patients only. Sequences from clinical cases are available in the Source Data file and GISAID EPI_SET_250516ap, where a proportion of positive samples from patients were sequenced as part of the genomic surveillance programme.

## Statistical analyses

Pearson correlations were performed to determine if there was any correspondence in SARS-CoV-2 content between Måby and Käppala (Fig. 1, N-gene/PMMoV ratio) or Käppala and Stockholm (Fig. 2, normalised SARS-CoV-2 weekly viral load). $P < 0.05$ were considered statistically significant. All the statistical analyses were conducted in Prism version 10 (GraphPad Software, CA, USA). Standard deviations ($s$) were calculated by Eq. (1) and the variance ($s^2$) using Eq. (2), where: ($x_i$) represents each data point, ($\mu$) is the mean of the data and ($N$) is the number of data points.

$$s = \sqrt{\frac{\sum(X_i - \mu)}{N}} \tag{1}$$

$$s^2 = \frac{\sum(X_i - \mu)}{N} \tag{2}$$

## Reporting summary

Further information on research design is available in the Nature Portfolio Reporting Summary linked to this article.

# Data availability

Data from wastewater surveillance is accessible at the European Nucleotide Archive ENA under Project: PRJEB61810. As for the occurrence data of COVID-19 clinical cases confirmed by PCR, it is publicly available through http://fohm-app.folkhalsomyndigheten.se/Folkhalsodata/pxweb/en/A_Folkhalsodata/A_Folkhalsodata_H_Sminet_covid19_testdata/PCRtest.px/. Source data are provided with this paper. Sequences from clinical cases are available in the source data file and GISAID EPI_SET_250516ap (https://doi.org/10.55876/gis8.250516ap). EPI_SET_250516ap is composed of 2,524 individual genome sequences and the collection dates range from 2023-01-02 to 2023-06-04. PRJEB61810: (fastq files from wastewater in Stockholm and Malmö, including aircraft and airport). PCR results: http://fohm-app.folkhalsomyndigheten.se/Folkhalsodata/pxweb/en/A_Folkhalsodata/A_Folkhalsodata_H_Sminet_covid19_testdata/PCRtest.px/. (for the data in this paper, select: Region: '01 Stockholm'; Kategori (category): 'Antal positiva' (number of positive cases); Vecka (week): 1–22; Åldersgrupp (age group): 'Alla åldrar' (all ages); Kön (sex): 'Totalt' (total); År (year): 2022)). Source data are provided with this paper.

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

## Acknowledgements

This work was funded by grants from the Swedish Government to the Public Health Agency of Sweden for assignments S2022/04841 and 05204-2022 (M.P.Z., C.B., N.L.M., H.B., C.S., Z.C.G.) and SciLifeLab, Pandemic Laboratory Preparedness (VC-2022-0066) (M.P.Z., I.O.A., A.K., L.W.H. and Z.C.). We acknowledge the funding support that made this work possible. The authors would like to thank Swedavia AB, Stockholm Arlanda Airport, the Käppala Association and Stockholm Vatten och Avfall for sample collection. We thank the Swedish regional infection control units and clinical microbiological laboratories from the Region Stockholm for their collaboration regarding the collection of clinical specimens and submission of whole genome sequence data. We thank colleagues at the Public Health Agency of Sweden especially Lena Dillner, Anna Risberg, Anette Hansen, Alma Brolund, and Stockholm Region. We also thank SciLifeLab, part of the National Genomics Infrastructure (NGI) Sweden. We extend our gratitude to researchers and their respective laboratories who have been instrumental in obtaining the clinical specimens globally. Their efforts in generating and sharing the genetic sequence and metadata via the GISAID Initiative have been invaluable to our research.

## Author contributions

M.P.Z. contributed to the conceptualisation, investigation, methodology, data curation, validation, formal analysis, visualisation and writing—original draft. C.B.: Data curation, formal analysis, methodology, project administration, resources, software, supervision, validation, visualisation. N.L.M.: Data curation, project administration, resources, software, supervision, validation, visualisation. I.O.A., A.K.: Investigation, methodology, formal analysis. H.B. and C.S.: Conceptualisation, investigation, funding acquisition. L.W.H.: Investigation, methodology, data curation, validation, formal analysis, software, visualisation. Z.C.: Conceptualisation, methodology, supervision, funding acquisition. All authors revised the manuscript and approved the final version.

## Funding

## Competing interests

The authors declare no competing interests.
