## [Peer Review file · Nature Communications]

Wastewater surveillance of SARS-CoV-2 from aircraft to citywide monitoring

Corresponding Author: Dr Zeynep Cetecioglu Gurol

Version 0:

Reviewer comments:

Reviewer #1

(Remarks to the Author)

General Comments:

This manuscript reports an interesting comparison of SARS-CoV-2 genomic variants detected in aircraft and airport wastewater with variants detected in municipal sewage and clinical cases in the same city. The study provides additional evidence for the growing consensus of wastewater surveillance's greater sensitivity and timeliness for identifying genomic variants than clinical surveillance. The detection of unique variants in aircraft wastewater not yet seen in the city demonstrates the feasibility of port-of-entry wastewater monitoring for identifying potential introductions of new variants to a population/geographic area. The manuscript would benefit from clarifying the description and organization of the methods, particularly concerning sample processing and pooling procedures.

Specific Comments:

L28: Avoid using contractions ("didn't"); also in Figure captions and the discussion.

L50: This likely has a short shelf life. Consider revising along the lines of "As of January 2024, WBE programs were active in at least 72 countries."

L108 – 111: I do not quite follow the modified concentration procedure described here. Am I correct in understanding that four separate samples were each concentrated, after which two concentrates were combined to create a pooled concentrate for nucleic acid extraction (two pooled concentrates total)? That is to say that the equivalent sample volume (ESV, see Crank et al. 2023, <https://doi.org/10.1021/acs.est.3c07968>) that was analyzed in each qPCR reaction did not increase, rather a potentially more representative pooled sample was analyzed? If so, I would argue it is misleading to describe this process as "increas[ing] the amount of virus". If I have misunderstood and the ESV was increased, please revise the method description to clarify how this was accomplished.

L124 – 125: How many qPCR technical replicates were analyzed for each sample and how were discrepancies in detection status between replicates handled?

L140 - 145: The pooling procedures are unclear. Is this saying that the purified RNA from two replicates for each sampling point was mixed together for a total volume of 25 mL purified RNA across all the sampling points each week? Were the airplane and city wastewater samples all pooled prior to sequencing, meaning that airplane wastewater variants could not be differentiated from the city wastewater variants? What is the 4.5 mL of wastewater from each inlet sample that is being pooled here then, which sounds like a pre-extraction procedure? Was a separate extraction conducted on wastewater pooled across all 6 inlets for sequencing, in addition to the separate extractions performed for each sampling location that were analyzed for viral load by RT-qPCR? Please clarify the description of sample handling, processing, and pooling, ideally condensing these descriptions into a single section earlier in the methods prior to describing nucleic acid extraction and molecular analyses, and with particular attention to which specific sample types, and at which processing stages, the samples were pooled. Please also provide a rationale for why each type of pooling was performed.

L168: "SARS-CoV-2 content" is ambiguous. Were these the quantitative "PMMoV factor"/" N-gene/PMMoV ratio" metrics that were being compared? Were Pearson correlations also used to compare the variants detected in wastewater to clinical

variants? Consider replacing “SARS-CoV-2 content” throughout with either the particular metric being compared (e.g., N-gene/PMMoV ratio in sample location A vs. sample location B) or another term that is more clearly quantitative, e.g. “normalized SARS-CoV-2 weekly viral load” (the PMMoV factor metric), the “SARS-CoV-2 relative abundance” (the N-gene/PMMoV ratio), or even perhaps “normalized SARS-CoV-2 quantity” to cover both metrics.

L191-192: The consideration of all three N gene assays and their comparison on the basis of detection limit should have been noted in the methods section, along with the criteria used to select the single primer set to use for the rest of the study. As reported, it's not clear from the results of this comparison why N3 was selected over N2.

L211: Can the authors comment on how confident we should be in the reliability of the lineages identified by Freya at “deepest lineage level available”? In contrast with essentially single-lineage clinical specimens, what should we make of fine-scale lineage differences identified by deconvolution of the highly mixed, likely degraded, and complex matrix of wastewater samples? As an example, is it possible that some of the 14 lineages identified solely in aircraft (Lines 247-249) may have been artifacts of the deconvolution approach rather than true lineages present in the sample?

L252 – 266: Might the authors comment on the practical value of detecting a greater diversity of variants in wastewater and detecting particular variants earlier in wastewater than in clinical cases? What would a public health agency actually do with this information? Presumably, many of these additional variants detected in wastewater are of low clinical relevance—they are not showing up in clinics. Some may be early indications of variants that will go on to be meaningful in the future, but what good is early detection if you don't know which of the unique-to-wastewater variants will become important until weeks later when they start being detected in clinical cases (at which point there is no longer an early warning advantage to wastewater). This of course touches on larger questions of the practical utility of wastewater surveillance more broadly, which is obviously beyond the scope of this study, but if the authors might be able to discuss some practical uses for data of the sort generated in this study, it would help make a more compelling case for the public health significance of the approaches described in this work.

Reviewer #2

(Remarks to the Author)

This study investigated SARS-CoV-2 genome sequences from wastewater collected at airport, aircraft, wastewater treatment plants, and Stockholm metropolitan areas to understand the relationships between these locations and the prevalence of circulating variants. Authors compared wastewater-derived sequences with those from clinical samples and found that while variants from aircrafts and airport were less prevalent in the city, wastewater monitoring consistently detected a broader range of variants than clinical testing, demonstrating its effectiveness for early variant detection.

The concept of using aircraft and airport wastewater to monitor for emerging SARS-CoV-2 variants has been explored in previous studies, which were not cited (e.g., Ahmed et al., 2022; Farkas et al., 2023; Tay et al., 2024). While this study provides valuable insights, it would benefit from a more comprehensive review of existing literature to highlight its unique contributions and potential limitations.

A significant challenge in attributing the source of circulating variants to air travelers is the difficulty in accounting for other modes of transportation, such as land and sea travel. Unless strict movement restrictions are in place, it is challenging to definitively conclude that air travel is the primary source of variants in the city. While this study may effectively rule out air travelers as a major source of prevalent variants, it might not fully address the broader question of variant importation. A more robust study design would be needed to definitively establish the role of air travelers in variant transmission.

Specific comments:

Abstract: this study only tested aircraft wastewater from flights originating from China, which presumably made up just a small portion of all inbound flights at the Arlanda Airport. This should be made clear in either the abstract or the title.

Line 36: PANGO lineage, rather than pangolin lineage

Line 85-86: a map may help readers understand the geographical relationship among the testing sites.

Line 108: more details should be provided here, why split four independent samples into two groups and mix them afterwards? Does this step increase the volume of wastewater before concentration from 40mL to 160mL?

Line 176-178: is the PMMoV concentration comparable to previous studies? Also, can a comparison between the PMMoV variability of wastewater and aircraft samples be provided?

Line 180-182: is there any data to show how they compare to the clinical cases during the same period?

Line 182-183: weren't four independent samples (160 mL in total) used for aircraft samples?

Line 186-187: do SARS-CoV-2 contents follow normal distribution?

Line 204-205: also because the corresponding population was different in each flight.

Line 211-212: is there an explanation why pooled samples had 229 more unique lineages than composite samples?

Line 250-251: how about the growth of their relative abundance?

Line 254-256: could it be due to the difference in sequencing methods and the possibility that some lineages were shed into wastewater by asymptomatic patients and these lineages do not pose clinical seriousness?

Figure 3 and 4c: sequencing data Kappala WWTP in week 5 seems to be missing, is there an explanation? Also, is there an explanation why aircraft and airport samples had the largest portion of undetermined lineages?

References:

Ahmed et al., 2022. Wastewater surveillance demonstrates high predictive value for COVID-19 infection on board repatriation flights to Australia. *Environmental International*, 158, 106938.

Farkas et al., 2023. Wastewater-based monitoring of SARS-CoV-2 at UK airports and its potential role in international public health surveillance. *PLOS Global Public Health*, 3(1), e0001346.

Tay et al., 2024. Usefulness of aircraft and airport wastewater for monitoring multiple pathogens including SARS-CoV-2 variants. *Journal of Travel Medicine*, 31(5), taae074.

Reviewer #3

(Remarks to the Author)

Reviewer #4

(Remarks to the Author)

Overview:

This manuscript describes a study that compares the SARS-CoV-2 lineages in wastewater from aircraft and airport sewage facilities with local population data to assess whether variants initially identified in the aircraft or airport spread into the metropolitan area (or vice versa). From January until May 2023, the authors collected weekly wastewater samples from five locations, including the aircraft, the airport terminal, a wastewater treatment plant (WWTP) serving the airport region and several other WWTPs serving the broader metropolitan area of Stockholm. Samples were 24-hour composite wastewater samples (except for the airport terminal, where grab samples were taken), and there were some gaps in sampling on the aircraft due to the cancellation of flights. The key results were that the SARS-CoV-2 identified in wastewater treatment plants reflected local infection rates and that variants initially found only on the aircraft and in the airport did not spread widely throughout the metropolitan area. The authors also highlighted that, compared to clinical tests, wastewater monitoring was more effective in the early detection of specific variants, and in the detection of a broader range of variants.

In general, I found numerous shortcomings of the study, which severely limit its potential significance. My comments are summarized below:

Related to the methods and the validity of the data and its interpretation:

The methodology is not described or presented in sufficient detail, leaving me with a lot of questions about what was actually done. This limits the quality of the data. For example, there is a lack of description of quality assurance and quality control measures used in the laboratory (e.g., positive controls, negative controls, matrix spikes, recovery controls, blanks, etc.). Not all appropriate controls have been included. In the supplemental materials document (lines 14-15), the authors describe that "Positive, negative, and cross-plate controls were the same as those described above", but nothing was described above about positive, negative, or cross plate controls. In the Results section, it is stated that "the PCR negative and positive controls performed as expected." For negative controls, this is clear (non-detects are expected). But it is less clear what is expected for positive controls (and this is further limited by my lack of understanding about what the positive controls were). Without the appropriate inclusion and analysis of positive and negative controls, the data may not be technically sound.

To begin, the description of the sample sites (lines 86-92) is a bit unclear. I cannot easily determine how many sites were actually sampled. It seems that more than 5 samples were collected each week, because the Bromma and Henriksdal WWTPs have several "inlets" each. It's unclear what is meant by "inlet" and assuming that inlets are referring to distinct sanitary sewer mains or trunks, it's unclear if and why the inlets were each sampled separately (or the authors meant on lines 91-92 that the six inlets were composited into a single sample).

Lines 99-100: specify the maximum holding time—what is meant by "within a day"? Does that mean within 24 hours? Or that they were processed the same day as they were collected? Or could it be that samples were collected in the morning of one day and then processed after "a day", like at the latest by the end of the next day (so up to 36 hours or so after sample

collection)? Since the samples were 24-hour composites, then it may have been close to 48 hours or even more until samples were concentrated and extracted (depending on what is meant by “a day”).

Lines 104-105: “Two independent replicates”—does this mean field duplicates (i.e., duplicate 24-hour composite samples were collected independently, then concentrated and extracted independently) or were they laboratory duplicates (i.e., the 24-hour composite samples were split into equal volumes, which were concentrated and extracted independently). Or does it mean that the extracted nucleic acids were analyzed independently using RT-qPCR (i.e., instrument replicates)?

I understand that tap water was used as a negative control, but were other typical RT-qPCR negative controls used? Like process control blanks, extraction blanks, no template controls, etc.?

Was the new concentration procedure evaluated against other established procedures to determine the efficiency of recovery, the accuracy, etc.?

Likewise, it's not very clear on lines 108-111, how the samples were processed. The authors state “four independent samples” so does that mean there were four autosamplers at each location (I'm guessing not), or that the composite sample volume was split into four 40 mL aliquots (i.e., laboratory processing duplicates)? Then, I assume that the concentrated volumes from two 40 mL aliquots were mixed (producing duplicate composite sample concentrates), which were each then extracted independently (producing duplicate nucleic acid extracts from four 40 mL aliquots of a single composite sample). This should be made very clear in the methods section so that the reader does not have to guess or assume what was actually done.

It would have been more clear to specify the type of RT-qPCR used earlier in the manuscript (e.g., SYBR, probe-based, etc.). I figured out it was SYBR from reading below in the methods section about analysis of melt curves, but it should have been stated more clearly, and it should have been justified (why SYBR instead of probe-based RT-qPCR?). Also, the justification for using the N3 primers instead of or in addition to N1 and/or N2 did not appear to be very strong to me. Most other wastewater surveillance studies have used N1 and N2 in addition to or instead of N3, and many have reported preference for N1 and N2 (often analyzed in duplex). The supplementary materials document describes the approach used to compare the three primer sets targeting the N gene, and Figure S1 does indicate that N3 produced higher concentrations than N1, but it also indicates that N2 produced higher concentrations than N1, and there does not seem to be any significant difference between N2 and N3 concentrations. Therefore, I find the justification for using N3 to be lacking (especially considering that it goes against the assays that have been used in numerous other studies).

The authors stated (lines 124-125) that reactions “were considered positive if the cycle threshold was less than 40 cycles with a single melting peak at the correct temperature.” This is not described correctly. Probably it was meant that the fluorescence crossed the established threshold before 40 cycles, but how was the threshold set? Was it set manually, or automatically (using software)? If software was used, which one? Was a sample considered positive if only one of two replicates amplified? How was the limit of detection determined? What about the limit of quantification? These should both be reported for all assays used. These are important details that must be considered and described in the methods section. Bustin et al. 2009 (DOI: 10.1373/clinchem.2008.112797) provide recommendations about the minimum information that should be provided when using qPCR or RT-qPCR, and this manuscript lacks some of this information.

For example, information about the standard curves is lacking (I see plasmids were used for PMMoV, but what about SARS-CoV-2 N3?). Also, what is meant by a “control sample of RNA from wastewater” that was used for reference and quality control? Presumably, this is RNA from a wastewater sample that tested positive for N3, but it's unclear how it can be used as a control, especially if matrix spikes or other types of inhibition controls were not performed.

Lines 116-117 and 127-131: There is not yet consensus in the literature for using PMMoV to normalize data. See Mazumder et al. 2022 (DOI: 10.1016/j.coesh.2022.100363) and Greenwald et al. 2021 (DOI: 10.1016/j.wroa.2021.100111) as two examples where multiple normalization factors were tested (including PMMoV, crAssphage, and others). Neither group found evidence that PMMoV normalization increased correlation with case numbers. In fact, one of them reported that crAssphage was more appropriate for normalization. These two studies are only examples. There are also many other studies that have examined normalization factors for wastewater surveillance.

For the calculations section (126-138), it is unclear how the “total SARS-CoV-2 detected in the Stockholm region” was calculated from various inlets. Were flow rates measured? Were loadings calculated? Were concentrations averaged? Unless you're measuring flow rates and calculating loadings, you're potentially getting a biased estimate of “total SARS-CoV-2” (especially considering the potential limitations of using PMMoV for normalization, as described in comment above). It also seems like the normalization methods used were inconsistent between sampling sites (lines 134-138), which presents problems when comparing between sites. In the analysis of the PMMoV results, the authors noted a greater variability in the concentrations of PMMoV in the aircraft samples compared to the WWTPs, and this was attributed to the “fluctuating faeces-to-water ratio as vacuum toilets have minimal water usage.” However, could it also have been attributed to the smaller population of individuals using the aircraft toilets, relative to the population being served by the WWTPs? PMMoV is a pepper pathogen, so the presence of PMMoV in human feces is dietary based, and if someone does not eat peppers, it would be logical to assume that their feces would not contain PMMoV. Since the population on each aircraft is much much smaller than the populations served by the WWTPs, if only a fraction of the population sheds PMMoV, it would logically be more likely to see Poisson-like variations in the number of people shedding PMMoV in aircrafts, and thus higher variability in the concentrations of PMMoV in aircraft wastewater.

My expertise in RNA metagenomic sequencing is more limited relative to my experience with the use of RT-qPCR, but I saw fewer issues with the description of the methods used for sequencing and data analysis, except that there did not appear to be any negative or positive controls.

All things considered, there are too many details lacking and limitations associated with the methods section to have confidence in the quality of the data. The interpretation of some of the results also lacked statistical rigor. As such, conclusions drawn are severely limited due to the lack of strong and scientifically sound evidence.

Regarding the analytical approach:

The authors used Pearson correlation analysis (line 167), but did not describe if the data were normally distributed. If data are not distributed normally, then Spearman's correlation analysis may be more appropriate. This is a limitation. Also, on lines 206-207, the authors noted a lack of notable differences when comparing Kåppala to the other WWTP inlets. Besides visually inspecting Figure S4, it does not appear that any statistical analysis was done to assess potential differences. The analytical approach is also severely limited by the lack of appropriate experimental controls, such as matrix spikes or RT-qPCR inhibition controls.

Conclusions:

In summary, the shortcomings described above severely limit the potential significance of the findings presented herein. This is especially the case, considering how many other studies have drawn similar conclusions over the past four years. For example, the conclusion that wastewater surveillance can detect "more variant diversity" and that it was "more effective ... in the early detection of specific variants" is not a novel finding, and the authors did not cite previous studies that have drawn similar conclusions. There are many studies that have drawn these conclusions, but here are just a few from 2022 as examples:

Vo et al. (2022). Use of wastewater surveillance for early detection of Alpha and Epsilon SARS-CoV-2 variants of concern and estimation of overall COVID-19 infection burden. *Science of The Total Environment*, 835, 155410.

Jahn et al. (2022). Early detection and surveillance of SARS-CoV-2 genomic variants in wastewater using COJAC. *Nature Microbiology*, 7(8), 1151-1160.

Karthikeyan et al. (2022). Wastewater sequencing reveals early cryptic SARS-CoV-2 variant transmission. *Nature*, 609(7925), 101-108.

The inclusion of aircraft wastewater is a component of this manuscript that has been less studied in previous works, but there have been other aircraft wastewater surveillance studies, and the discussion section lacks a comparison of this study's results to other studies of aircraft wastewater surveillance. Here are a few examples of these studies.

Ahmed et al. (2020). Detection of SARS-CoV-2 RNA in commercial passenger aircraft and cruise ship wastewater: a surveillance tool for assessing the presence of COVID-19 infected travellers. *Journal of Travel Medicine*, 27(5), 1-11.

Morfino et al. (2023). Notes from the field: aircraft wastewater surveillance for early detection of SARS-CoV-2 variants—John F. Kennedy International Airport, New York City, August–September 2022. *MMWR. Morbidity and Mortality Weekly Report*, 72.

Jones et al. (2023). Suitability of aircraft wastewater for pathogen detection and public health surveillance. *Science of The Total Environment*, 856, 159162.

All things considered, the results from this study do not significantly advance understanding about the science of wastewater surveillance and lack the potential to move the field forward significantly.

Version 1:

Reviewer comments:

Reviewer #1

(Remarks to the Author)

The authors have addressed many of the issues raised in the previous review and the revised manuscript is much improved. I have noted a couple minor points below.

L148 - 149: Thank you for clarifying how replicates were analyzed. Please specify what quantity this standard deviation was calculated for. Was it the standard deviation of the Ct value between replicates? log₁₀ copies/uL? etc.

L154 - 157: Thank you for including this LOD and LOQ information. I assume these values were determined in the authors' previous work. Because the definitions of LOD and LOQ vary widely between studies, please either define them here and

describe how they were determined for these assays or explicitly direct the reader to the previous paper(s) where that information is provided. Additionally, it is not clear whether these limits are in terms of μL of reaction volume or μL or RNA template volume. Since a new concentration approach was implemented partway through the study with the explicit goal of improving detection sensitivity, the authors might consider converting these limits into copies/mL wastewater (or similar) for the two concentration approaches to provide a sense of the expected sensitivity improvements from the new concentration method.

Reviewer #2

(Remarks to the Author)

I greatly appreciate authors' intensive efforts to revise this manuscript according to reviewers' comments. I think the presentation of the manuscript has been significantly improved, but some minor changes are still required. One of the conclusions reached by authors is that wastewater monitoring is more effective than clinical testing in early detection of specific variants, with notable delays observed in the clinical surveillance (L28-29). However, when targeting specific variants, like Variants Under Monitoring (VUMs), Variants of Concern (VOCs), and Variants of Interest (VOIs) in wastewater, RT-qPCR rather than next-generation sequencing is sufficient for viral sequence detection. This point should be more explicitly described in the text, including abstract. Regarding the required number of wastewater treatment plants (WWTPs) to perform city-wide infection assessment, this is highly dependent on local conditions, including the size of the sewer-shed (population coverage) and the type of sewer system (separated or combined). In some cases, a single large WWTP covering 70-80% of the population in a large city may be sufficient for monitoring. More detailed discussion of this issue is necessary.

Reviewer #3

(Remarks to the Author)

Reviewer #4

(Remarks to the Author)

The revised manuscript is an improvement over the original submission. However, there are still some shortcomings. Below are my comments.

Both original reviewers were originally confused about the methodology, and several limitations were pointed out in the original review. Most of the confusion has been cleared up in the revised version of the manuscript, but there are still a few items that could use more clarification and there is one major limitation due to an omitted control.

The authors added more details to this part of the methods section and clarified that the equivalent sample volumes indeed increased due to concentration using a composited sample based on four independent replicates. On line 115, it should be stated that the manufacturer's recommendations were followed, with some exceptions, because on line 124, the "new concentration procedure" (which is actually just a four-sample composite) appears to be a modification from the manufacturer's recommended protocol. This is also the only reason why the ESV is different (because instead of processing 500 μL as recommended in the protocol, a composited 1000 μL was processed). Also, instead of describing it as a "new concentration procedure", I think it'd make more sense to call it a spatial composite of aircraft wastewater (which is probably highly heterogeneous in nature). So, by collecting 4 replicates and mixing them together, it's essentially a composite sample. The authors should describe how the 4 replicates were selected (Did they all come from the same location in the wastewater tank? Or were some from the bottom of the tank and others from the top, etc.?).

It's still not clear why N3 was selected over N2 or N1 and why SYBR green was selected over TaqMan. The authors stated on lines 258-259 that the comparison between SYBR green and TaqMan methods "revealed no clear differences between the two methods," however after reviewing the supplementary information, there were huge differences between the CT values obtained using SYBR green vs. TaqMan, with some samples showing more than 5 CT units difference! This would equate to more than an order of magnitude difference in the concentrations. Based on those results, there is no valid justification for using SYBR green over TaqMan, especially given that TaqMan is a more specific method than SYBR green. Regarding the selection of N3 over the other targets, the authors cited one paper that reported this to be a better target but has ignored much of the literature that has used N1 and/or N2 over N3 (which constitutes the majority of studies).

The results of the standard curves are not disclosed. Things like the slope, intercept, R^2 value, and most importantly, the efficiency (which should be between 90% and 110%). This is important to report especially since the SARS-CoV-2 assays often had poor efficiencies in other published studies (sometimes they were much greater than 110%).

In general, part of the issue with the confusion about the methods and the lack of confidence in the results is that the authors do not follow the widely-cited MIQE guidelines for the publication of results from qPCR (Bustin et al. 2009; doi: 10.1373/clinchem.2008.112797). This minimum information should be disclosed in all publications that use qPCR,

according to that study, which has thousands of citations.

In the revised methods section and supporting information document, the authors added many details about many of the normal controls used during qPCR (nuclease free water in qPCRs, which is commonly called a “no template control”, as well as details about the plasmids used as the positive control to construct the standard curve), however many other typical qPCR controls are missing or were done in a non-conventional way. For example, the tap water used as a negative control is not a conventional way to run a negative control. The cross-plate control is a nice QA/QC addition, but it is unclear what the authors did with the information from these controls. Presumably the CT values were not vastly different from plate to plate, but what was the range of differences observed in the cross-plate controls? That should be reported, at least in the Supporting Information document. Many typical controls were omitted. Typically, field blanks, process blanks, and extraction blanks, consisting of reagent water, MilliQ water, or nuclease free water, are analyzed alongside samples to determine if there was contamination in the field, during sample concentration, or during nucleic acid extraction. The authors stated that the tap water negative controls “performed as expected” and presumably that means there was no amplification at all, but this was not explicitly stated.

The most important control that is missing in this study is the PCR inhibition control. Wastewater has numerous potential PCR inhibitors which can change from day to day and sample to sample, potentially introducing major biases into the interpreted results due to PCR inhibition (which would make the concentration appear to be lower than it actually might have been). PCR inhibition is most commonly assessed using the TaqMan method with an internal amplification control (IAC), which is an amplicon with the same primer pair but a different probe sequence that a known quantity of which is added to a qPCR replicate for every sample. Or, alternatively, an additional replicate of every sample is analyzed with a known quantity of standard (e.g., plasmids) added to the qPCR reaction well (or some people analyze an additional replicate but with a dilution of the template RNA). If the C_q value does not change as expected with the plasmid spike or the diluted template RNA, then it is a sign of inhibition. Without some type of PCR inhibition controls, the reliability of the data is limited.

DISCUSSION:

The revised manuscript had minimal revisions to the discussion section. A previous reviewer brought up the observation that the authors did not discuss their results within the context of the findings from previously published aircraft wastewater surveillance studies. These previous studies are cited (in the introduction), but there is no discussion about how the results from the present study advance the knowledge of what has been reported previously, including in studies of aircraft wastewater. The revised manuscript still states (lines 307-309) that “the influence of SARS-CoV-2 contents and variants of aircraft and airport regions on urban areas remains unexplored.” Ahmed et al. (2020) detected the virus in aircraft wastewater five years ago. How did this study advance their work? Morfino et al. (2023) used aircraft wastewater surveillance to identify SARS-CoV-2 variants. What was different about the findings of this study compared to that one? Were the conclusions similar? Different? Jones et al. (2023) did a thorough assessment of the utility of aircraft wastewater surveillance, quantifying toilet use/behavior on flights of different durations. How were the findings from that study incorporated into the present study? Did the authors consider the duration of the flights? Why or why not, given the findings from Jones et al. (2023). I’m not suggesting that the authors respond to each of these questions posed here, these are just examples of how one might discuss the results of a study in comparison/contrast to previously published papers on the topic. My point is that the discussion section of this manuscript is limited in that it does not relate the findings to any of the previously published findings from studies of aircraft/airport wastewater surveillance.

CONCLUSIONS:

The revised manuscript had no revisions to the conclusions section. A previous reviewer reported concerns about the lack of novelty in the conclusions. Papers published by Nature Communications journal are typically very high impact studies that represent “important advances of significance” to specialists within a particular field. The conclusions of the revised manuscript fall short of this expectation. The data from the metagenomics analysis indeed showed that wastewater surveillance can detect “more variant diversity” earlier than clinical surveillance. It is important to study variants of concern and to have a way to make early predictions about their dissemination. I don’t disagree that this study showed that wastewater surveillance has the potential to provide this important information earlier than clinical surveillance, and this would have been a very impactful and novel finding in 2020 or 2021. However, at this point (now more than 5 years after the start of the pandemic), the conclusion is only confirming the conclusions from a plethora of previous studies, many of which were published years ago. The authors indeed cited some of these previous studies, but adding more citations to the manuscript does not change this fact that the conclusions of this study do not represent an “important new advance of significance” to the field.

Version 2:

Reviewer comments:

Reviewer #2

(Remarks to the Author)

I confirmed that all my comments have been appropriately addressed.

General Response

Thank you for your valuable feedback and insightful comments. We greatly appreciate the thorough review, which has sparked engaging discussions among the authors. Your input has been taken seriously, serving as essential guidelines for revising our manuscript.

Thank you once again for your valuable insights.

Point-by-point responses to the comments.

REVIEWER COMMENTS

Reviewer #1 (Remarks to the Author):

General Comments:

This manuscript reports an interesting comparison of SARS-CoV-2 genomic variants detected in aircraft and airport wastewater with variants detected in municipal sewage and clinical cases in the same city. The study provides additional evidence for the growing consensus of wastewater surveillance's greater sensitivity and timeliness for identifying genomic variants than clinical surveillance. The detection of unique variants in aircraft wastewater not yet seen in the city demonstrates the feasibility of port-of-entry wastewater monitoring for identifying potential introductions of new variants to a population/geographic area. The manuscript would benefit from clarifying the description and organization of the methods, particularly concerning sample processing and pooling procedures.

General Response

Thank you for your insightful comment on the manuscript. Your recognition of the study's comparison of SARS-CoV-2 genomic variants across different wastewater sources and clinical cases is appreciated. We agree that the findings underscore the enhanced sensitivity and timeliness of wastewater surveillance in identifying genomic variants.

We acknowledge your suggestion regarding the methods section. We clarified the description of the sample processing and pooling procedures to ensure the methodology is comprehensively understood.

Specific Comments:

L28: Avoid using contractions (“didn’t”); also in Figure captions and the discussion.

A. Thank you for your observation. We have modified the text according.

L50: This likely has a short shelf life. Consider revising along the lines of “As of January 2024, WBE programs were active in at least 72 countries.”

A. Thank you for your suggestion. We understand that the original statement might quickly become outdated. To address this, we will revise it to: “As of January 2024, WBE programs

were active in at least 72 countries.” This provides a clear timeframe and ensures the information remains relevant and precise.

L108 – 111: I do not quite follow the modified concentration procedure described here. Am I correct in understanding that four separate samples were each concentrated, after which two concentrates were combined to create a pooled concentrate for nucleic acid extraction (two pooled concentrates total)? That is to say that the equivalent sample volume (ESV, see Crank et al. 2023, <https://doi.org/10.1021/acs.est.3c07968>) that was analyzed in each qPCR reaction did not increase, rather a potentially more representative pooled sample was analyzed? If so, I would argue it is misleading to describe this process as “increas[ing] the amount of virus”. If I have misunderstood and the ESV was increased, please revise the method description to clarify how this was accomplished.

A. Thank you for your comment. Indeed, the ESV increased from 2.5×10^{-3} L to 5×10^{-3} L. The EVS equation is:

$$ESV (L) = V_{\text{processed}} (L) \times \frac{V_{\text{extraction}} (\mu\text{L})}{V_{\text{concentrate}} (\mu\text{L})} \times \frac{V_{\text{template,RT}} (\mu\text{L})}{V_{\text{elution}} (\mu\text{L})} \times \frac{V_{\text{template,PCR}} (\mu\text{L})}{V_{\text{cDNA}} (\mu\text{L})}$$

In our case, we used a one-step reverse transcription qPCR supermix. Therefore, $V_{\text{template,RT}}$, $V_{\text{template,PCR}}$ and V_{cDNA} are the same. The equation will be:

$$ESV(L) = V_{\text{processed}}(L) \times \frac{V_{\text{extraction}}(\mu\text{L})}{V_{\text{concentrate}}(\mu\text{L})} \times \frac{V_{\text{template,PCR}}(\mu\text{L})}{V_{\text{elution}}(\mu\text{L})}$$

Normal samples:

$$ESV(L) = 40 \times 10^{-3} (L) \times \frac{500(\mu\text{L})}{500(\mu\text{L})} \times \frac{5(\mu\text{L})}{80(\mu\text{L})} = 2.5 \times 10^{-3} \text{ L}$$

Aircraft samples:

$$ESV(L) = 80 \times 10^{-3} (L) \times \frac{1000(\mu\text{L})}{1000(\mu\text{L})} \times \frac{5(\mu\text{L})}{80(\mu\text{L})} = 5 \times 10^{-3} \text{ L}$$

We have revised the method description to clarify the procedure. Section: Wastewater concentration and RNA extraction in Methods.

L124 – 125: How many qPCR technical replicates were analyzed for each sample and how were discrepancies in detection status between replicates handled?

A: Thank you for the observation. The following text was added to material and methods: “Two independent technical replicates were measured for each biological replicate and control sample (positive, negative, and cross-plate controls) using qPCR. Results were accepted if the standard deviation between the technical replicates was less than 0.5.”

L140 - 145: The pooling procedures are unclear. Is this saying that the purified RNA from two replicates for each sampling point was mixed together for a total volume of 25 mL purified RNA across all the sampling points each week? Were the airplane and city wastewater samples all pooled prior to sequencing, meaning that airplane wastewater variants could not be differentiated from the city wastewater variants? What is the 4.5 mL of wastewater from each inlet sample that is being pooled here then, which sounds like a pre-extraction procedure? Was a separate extraction conducted on wastewater pooled across all 6 inlets for sequencing, in

addition to the separate extractions performed for each sampling location that were analyzed for viral load by RT-qPCR? Please clarify the description of sample handling, processing, and pooling, ideally condensing these descriptions into a single section earlier in the methods prior to describing nucleic acid extraction and molecular analyses, and with particular attention to which specific sample types, and at which processing stages, the samples were pooled. Please also provide a rationale for why each type of pooling was performed.

A: Thank you for your comment. We made a significant mistake with the units, which caused the misunderstanding. We meant 25 μL and 4.5 μL , not milliliters. The section has been carefully revised and improved as follows:

“After the wastewater concentration and RNA extraction steps, 25 μL of purified RNA was shipped to the Uppsala Genome Center (Science for Life Laboratory, Dept. of Immunology, Genetics and Pathology, Uppsala University) for sequencing. The two biological replicates were mixed in equal amounts (12.5 μL) and sent for sequencing per sampling point. These sampling points included the six inlets of the WWTPs (Henriksdal, Sickla, Hässelby, Järva, Riksby, and Käppala), aircraft, Arlanda Terminal 5, and Måby (airport region) samples. For Stockholm composite samples, 4.5 μL of each WWTP inlet were pooled, resulting in a volume of 27 μL of sample per week. For weeks 7, 8, and 9, the mixed sample from Stockholm was not sequenced, and data was pooled in silico. Additionally, in silico pools including all six inlets, or five inlets excluding Käppala, were created when needed and specified in the results.”

L168: “SARS-CoV-2 content” is ambiguous. Were these the quantitative “PMMoV factor”/“N-gene/PMMoV ratio” metrics that were being compared? Were Pearson correlations also used to compare the variants detected in wastewater to clinical variants? Consider replacing “SARS-CoV-2 content” throughout with either the particular metric being compared (e.g., N-gene/PMMoV ratio in sample location A vs. sample location B) or another term that is more clearly quantitative, e.g. “normalized SARS-CoV-2 weekly viral load” (the PMMoV factor metric), the “SARS-CoV-2 relative abundance” (the N-gene/PMMoV ratio), or even perhaps “normalized SARS-CoV-2 quantity” to cover both metrics.

A: Thank you for your suggestion. We have revised the manuscript accordingly and replaced “SARS-CoV-2 content” with more appropriate terminology when necessary. For your reference see line 225, 236, 239, 241.

L191-192: The consideration of all three N gene assays and their comparison on the basis of detection limit should have been noted in the methods section, along with the criteria used to select the single primer set to use for the rest of the study. As reported, it’s not clear from the results of this comparison why N3 was selected over N2.

A: Thank you for your valuable comment. The following paragraph was added to the Method section: *“SARS-CoV-2 concentrations in aircraft samples were near the detection limit for several weeks during the monitoring period. For this reason, we evaluated whether the detection limit could be improved using TaqMan method or if there was any difference among the well-known primers targeting the N-gene (N1, N2 and N3 primers). The methodology and results are presented in supplementary material.”*

In addition, we incorporated a motivation why N3 was the preferred primer:

“The N3 primers were preferred over N2 due to evidence from another study indicating that N3 exhibited superior performance and better detection limits with stool samples.”

Reference: Lu X, Wang L, Sakthivel SK, *et al.* US CDC real-time reverse transcription PCR panel for detection of severe acute respiratory syndrome Coronavirus 2. *Emerg Infect Dis* 2020; 26. DOI:10.3201/eid2608.201246.

L211: Can the authors comment on how confident we should be in the reliability of the lineages identified by Freya at “deepest lineage level available”? In contrast with essentially single-lineage clinical specimens, what should we make of fine-scale lineage differences identified by deconvolution of the highly mixed, likely degraded, and complex matrix of wastewater samples? As an example, is it possible that some of the 14 lineages identified solely in aircraft (Lines 247-249) may have been artifacts of the deconvolution approach rather than true lineages present in the sample?

A. The pipeline Freyja is conservative in that it doesn't always attribute lineages to the fullest; instead, if there is low coverage or low depth, lineages are grouped at a higher order level, e.g. “XBB.1-like”, if a partial assignment could be made, or “Misc”, if the assignment already at the highest level is unclear. Together with the settings for depth and coverage that we used, this gives us good confidence that the lineages that do get full assignment. Indeed, the aircraft samples yielded a much higher proportion of these not-fully elucidated lineages: 35% vs. 4.3% on average from the other samples. We have clarified this in the methods*, lines 204-206, added this information to the results** section, lines 274-276, and included it in the discussion#, lines 320-322.

* “For statistical analyses, only variants that could be fully assigned by Freyja, without the ambiguity markers “Misc” or “-like” were considered”

** “Additionally, the data from the aircrafts, having lower coverage and depth, was more often not fully assigned by Freyja, receiving a truncated lineage. This affected 35% of the total sequence variants in the aircraft data, in comparison to 4.3% on average for the other sites.”

“This is likely a low estimation, as the low coverage of the aircraft samples led to more ambiguous assignments, which were discarded from our analyses.”

L252 – 266: Might the authors comment on the practical value of detecting a greater diversity of variants in wastewater and detecting particular variants earlier in wastewater than in clinical cases? What would a public health agency actually do with this information? Presumably, many of these additional variants detected in wastewater are of low clinical relevance—they are not showing up in clinics. Some may be early indications of variants that will go on to be meaningful in the future, but what good is early detection if you don't know which of the unique-to-wastewater variants will become important until weeks later when they start being detected in clinical cases (at which point there is no longer an early warning advantage to wastewater). This of course touches on larger questions of the practical utility of wastewater surveillance more broadly, which is obviously beyond the scope of this study, but if the authors might be able to discuss some practical uses for data of the sort generated in this study, it would help make a more compelling case for the public health significance of the approaches described in this work.

A Thank you for your considerations. As seen during several phases of the pandemic, new VOCs or VUMs have been detected thanks to thorough surveillance programs and intensive clinical sampling globally. The question, whether new variants already have spread globally is usually dependent on the continuous sampling efforts in every country and requires considerable resources. WBE offers the possibility to improve cost-effectiveness by screening a comparably low number of wastewater samples that cover a much higher proportion of the population and providing similar information.

Another aspect is that since sequencing of clinical samples is declining globally, the knowledge of which variants circulate and the resolution of that information decline. Here, WBE offers the possibility to establish cost-effective and continuous baseline data while keeping patient sample sequencing for situations where increased focus on individual cases is needed such as during local outbreaks.

We addressed these aspects in more detail in the section you mentioned.

Reviewer #2 (Remarks to the Author):

This study investigated SARS-CoV-2 genome sequences from wastewater collected at airport, aircraft, wastewater treatment plants, and Stockholm metropolitan areas to understand the relationships between these locations and the prevalence of circulating variants. Authors compared wastewater-derived sequences with those from clinical samples and found that while variants from aircrafts and airport were less prevalent in the city, wastewater monitoring consistently detected a broader range of variants than clinical testing, demonstrating its effectiveness for early variant detection.

The concept of using aircraft and airport wastewater to monitor for emerging SARS-CoV-2 variants has been explored in previous studies, which were not cited (e.g., Ahmed et al., 2022; Farkas et al., 2023; Tay et al., 2024). While this study provides valuable insights, it would benefit from a more comprehensive review of existing literature to highlight its unique contributions and potential limitations.

A significant challenge in attributing the source of circulating variants to air travellers is the difficulty in accounting for other modes of transportation, such as land and sea travel. Unless strict movement restrictions are in place, it is challenging to definitively conclude that air travel is the primary source of variants in the city. While this study may effectively rule out air travelers as a major source of prevalent variants, it might not fully address the broader question of variant importation. A more robust study design would be needed to definitively establish the role of air travelers in variant transmission.

General Response

Thank you for your insightful comments on our study. We appreciate your recognition of the value in using wastewater monitoring for early detection of SARS-CoV-2 variants.

We have revised the manuscript to include additional literature, such as Ahmed et al., 2022; Farkas et al., 2023; and Tay et al., 2024, to further emphasize our study's unique contributions.

Regarding the challenge of attributing the source of circulating variants, we acknowledge the complexity of accounting for other modes of transportation, such as land and sea travel. Our study aimed to rule out variant importation specifically from China and more broadly at Arlanda Airport. To achieve this, we targeted aircraft arriving from China and monitored the Måby station, which collects all wastewater within the airport region, including from all the aircraft arriving at Arlanda, hotels, airport facilities, administrative offices, and other areas.

By focusing on these specific and broad sources, we aimed to establish a clearer understanding of the role of air travelers from China and other regions in variant transmission within the population. This study evaluated SARS-CoV-2 contents and variants, determining lineages in wastewater from aircraft lavatories and Arlanda's airport sewage facilities and compared these

with local population data to assess transmission of variants between travellers and local population.

Specific comments:

Abstract: this study only tested aircraft wastewater from flights originating from China, which presumably made up just a small portion of all inbound flights at the Arlanda Airport. This should be made clear in either the abstract or the title.

A: Thank for your comment. We have clarified this in the abstract: *“Variants initially detected in the aircraft arriving from China and Arlanda airport region did not spread widely during the study period.”*

Line 36: PANGO lineage, rather than pangolin lineage

A. Thank you for the observation. It has been corrected.

Line 85-86: a map may help readers understand the geographical relationship among the testing sites.

A. Thank you for the suggestion. We have added two maps, Figure S1 and Figure S2, in the supplementary material.

Line 108: more details should be provided here, why split four independent samples into two groups and mix them afterwards? Does this step increase the volume of wastewater before concentration from 40mL to 160mL?

A. We have modified the paragraph for better understanding, as follows:

“Due to the relatively low SARS-CoV-2 content in aircraft samples, a new concentration procedure was implemented from week 5 onwards to enhance the likelihood of positive detection and increasing the amount of virus before RT-qPCR analysis. Four independent wastewater samples (40 mL each) were concentrated to 500 μ L using the Promega kit. Each 500 μ L concentrate was then split into two groups and combined to obtain approximately 1000 μ L. Subsequently, the combined samples were used for RNA extraction as previously described, resulting in two independent replicates of 80 μ L.”

Line 176-178: is the PMMoV concentration comparable to previous studies? Also, can a comparison between the PMMoV variability of wastewater and aircraft samples be provided?

A. Yes, they are comparable. For reference, a paper on PMMoV detection reported concentrations ranging from (10^5) to (10^8) gene copies/L in wastewater (Reference: Kitajima, M., Sassi, H.P. & Torrey, J.R. Pepper mild mottle virus as a water quality indicator. *npj Clean Water* **1**, 19 (2018). <https://doi.org/10.1038/s41545-018-0019-5>)

To better explain the variability of the samples, Ct values and gene copies per liter were included. The following information has been added:

“The average PMMoV Ct value across all inlets of the WWTPs in Stockholm is 23.58 ± 0.99 and a variance of 0.98 ($1.1 \times 10^7 \pm 6.4 \times 10^6$ gene copies/L and variance 4.1×10^{10} gene copies/L). However, aircraft samples showed increased PMMoV variability due to the fluctuating faeces-to-water ratio as vacuum toilets have minimal water usage, with an average PMMoV Ct value of 23.65 ± 3.07 and a variance of 9.43 ($2.4 \times 10^7 \pm 5.8 \times 10^7$ gene copies/L and variance 3.3×10^{15} gene copies/L).”

Line 180-182: is there any data to show how they compare to the clinical cases during the same period?

A. We do not have clinical data from aircraft, as travellers were only required to present a negative PCR certificate. However, we have made a clinical comparison for Stockholm area; for reference, see Figures 3 and 4.

Line 182-183: weren't four independent samples (160 mL in total) used for aircraft samples?

A. Four samples were concentrated and then divided into two groups. These groups were subsequently combined, resulting in a total of 80 ml of starting material, doubling the initial volume of 40 ml. We have enhanced the description in the Materials and Methods section to provide a clearer explanation of the procedure, as follow: *“Four independent wastewater samples (40 mL each) were concentrated to 500 μ L using the Promega kit. Each 500 μ L concentrate was then split into two groups and combined to obtain approximately 1000 μ L. Subsequently, the combined samples were used for RNA extraction, resulting in two independent biological replicates of 80 μ L.”*

Line 186-187: do SARS-CoV-2 contents follow normal distribution?

A. Based on the Shapiro-Wilk test, the data for Stockholm and K ppala follows a normal distribution. Therefore, Pearson correlations are appropriate for this analysis.

Line 204-205: also because the corresponding population was different in each flight.

A. Thank you for the suggestion, we have added the sentence in the manuscript.

Line 211-212: is there an explanation why pooled samples had 229 more unique lineages than composite samples?

A. We believe this could be due to a difference in sequencing depth. The composite samples were sequenced to the same depth as single samples, which would result in lower depth per variant. In contrast, the pooled samples are the sum of each sample, amounting to 5x deeper coverage. This has been added to that sentence, line 283: *“probably as a result of sequencing depth;”*

Line 250-251: how about the growth of their relative abundance?

A. These 14 lineages in particular were only ever detected in the aircraft; we have amended this sentence for clarity. An additional 9 lineages were found in the aircraft before being observed in other sites. We explain this more clearly in the discussion as follows: *“An additional 9 lineages were detected first in the aircraft, but they never accounted for more than 4.4% of any other location and time-point, remaining undetected in most samples.”*

Line 254-256: could it be due to the difference in sequencing methods and the possibility that some lineages were shed into wastewater by asymptomatic patients and these lineages do not pose clinical seriousness?

A. We agree, and have added a sentence to this effect in the discussion: *“Most of these variants are likely shed by asymptomatic carriers or those with mild symptoms, but this still proves that a VOC with international spreading would likely be detected earlier in wastewater than in the clinics.”*

Figure 3 and 4c: sequencing data Kappala WWTP in week 5 seems to be missing, is there an explanation? Also, is there an explanation why aircraft and airport samples had the largest portion of undetermined lineages?

Unfortunately, the read alignment file for that sample (week5-Käppala WWTP) was corrupted before we had a chance to back it up, leading to data loss. Therefore, we could not include in the manuscript.

The airport and aircraft samples had the lowest biomass, which therefore yielded a lower coverage of the viral genome. Since lineages are only assigned when there is sufficient coverage and depth, it was expected that these samples would have a higher proportion of undetermined lineages. This is now mentioned both in the methods* (line 205-208) and discussion# (lines 321-323):

* *“For statistical analyses, only variants that could be fully assigned by Freyja, without the ambiguity markers “Misc” or “-like” were considered”*

“This is likely a low estimation, as the low coverage of the aircraft samples led to more ambiguous assignments, which were discarded from our analyses.”

Reviewer #3 (Remarks to the Author):

Reviewer #4 (Remarks to the Author):

Overview:

This manuscript describes a study that compares the SARS-CoV-2 lineages in wastewater from aircraft and airport sewage facilities with local population data to assess whether variants initially identified in the aircraft or airport spread into the metropolitan area (or vice versa). From January until May 2023, the authors collected weekly wastewater samples from five locations, including the aircraft, the airport terminal, a wastewater treatment plant (WWTP) serving the airport region and several other WWTPs serving the broader metropolitan area of Stockholm. Samples were 24-hour composite wastewater samples (except for the airport terminal, where grab samples were taken), and there were some gaps in sampling on the aircraft due to the cancellation of flights. The key results were that the SARS-CoV-2 identified in wastewater treatment plants reflected local infection rates and that variants initially found only on the aircraft and in the airport did not spread widely throughout the metropolitan area. The authors also highlighted that, compared to clinical tests, wastewater monitoring was more effective in the early detection of specific variants, and in the detection of a broader range of variants.

In general, I found numerous shortcomings of the study, which severely limit its potential significance. My comments are summarized below:

Response to the specific comments:

Comment 1:

Related to the methods and the validity of the data and its interpretation:

The methodology is not described or presented in sufficient detail, leaving me with a lot of questions about what was actually done. This limits the quality of the data. For example, there is a lack of description of quality assurance and quality control measures used in the laboratory

(e.g., positive controls, negative controls, matrix spikes, recovery controls, blanks, etc.). Not all appropriate controls have been included. In the supplemental materials document (lines 14-15), the authors describe that “Positive, negative, and cross-plate controls were the same as those described above”, but nothing was described above about positive, negative, or cross plate controls. In the Results section, it is stated that “the PCR negative and positive controls performed as expected.” For negative controls, this is clear (non-detects are expected). But it is less clear what is expected for positive controls (and this is further limited by my lack of understanding about what the positive controls were). Without the appropriate inclusion and analysis of positive and negative controls, the data may not be technically sound.

A. Thank you for your comment. The information about negative, positive controls and cross-plate controls were mentioned in the submitted manuscript. We have improved the information about the controls.

Negative control for wastewater concentration and RNA extraction: Line 105 (submitted version): “tap water was used as a negative control”. We have improved the sentence as follows: “*Two tap water samples were used as negative controls in each extraction set (extraction of 16 samples).*”

Negative control for qPCR analysis: Line 117-118 (submitted version): “Nuclease-free water and RNA extracted from tap water were included as negative controls for all qPCR reactions.”

Positive controls for qPCR analysis: Line 118-121 (submitted version): “SARS-CoV-2 DNA (2019-nCoV_N_Positive Control, IDT, Cat. 10006625), and a constructed plasmid containing the appropriate target for PMMoV (IDT, custom MiniGene 25-500 bp) were used as positive controls and to create the standard curves.

Positive controls for qPCR analysis and quality control: Line 121-123 (submitted version). We have improved the sentence as follows: “*A control sample of RNA from wastewater (cross plate controls) with known SARS-CoV-2 and PMMoV concentrations was used in each qPCR analysis for reference and quality control, stored at -80°C, and used for up to eight weeks.*”

Supplementary material: We have improved the description of the positive, negative and quality controls. “*Two tap water samples were used as negative controls in each extraction set (extraction of 16 samples) and subsequently analysed by qPCR. Nuclease-free water was also used as negative control for qPCR analysis. SARS-CoV-2 DNA (2019-nCoV_N_Positive Control, IDT, Cat. 10006625), and a constructed plasmid containing the appropriate target for PMMoV (IDT, custom MiniGene 25-500 bp) were used as positive controls and to create the standard curves. Cross-plate controls with known SARS-CoV-2 and PMMoV concentrations were used in each qPCR analysis for reference and quality control.*”

Comment 2:

To begin, the description of the sample sites (lines 86-92) is a bit unclear. I cannot easily determine how many sites were actually sampled. It seems that more than 5 samples were collected each week, because the Bromma and Henriksdal WWTPs have several “inlets” each. It’s unclear what is meant by “inlet” and assuming that inlets are referring to distinct sanitary sewer mains or trunks, it’s unclear if and why the inlets were each sampled separately (or the authors meant on lines 91-92 that the six inlets were composited into a single sample).

A. Thank you for your observation. The inlets from the WWTPs were sampled separately because they originate from different areas in Stockholm. The presence of automatic samplers in each inlet allows us to take flow-compensated samples from each one. We have been monitoring SARS-CoV-2 in this manner for over three years, as it provides an overview of the COVID-19 infection in each area. Additionally, it allows us to integrate the data from each inlet to determine the total SARS-CoV-2 concentration for Stockholm.

We appreciate the reviewer's comment and have enhanced the description of the sampled regions in the revised manuscript. To aid in understanding the sampling sites, we have included maps in the supplementary material, Figure S1 and S2. The improved description of the sampling sites is the following:

"From January to May 2023, weekly wastewater samples were collected from aircraft, Arlanda Airport Terminal 5, Måby station (covering the airport region), Käppala WWTP, Bromma WWTP, and Henriksdal WWTP (Figure S1 and S2, supplementary material). Stockholm city includes the three main WWTPs: Bromma (18% of the population), Henriksdal (41%), and Käppala (33%). Bromma WWTP has three inlets: Hässelby, Riksby and Järva, while Henriksdal WWTP has two inlets: Sickla and Henriksdal. All airport wastewater goes to Käppala (Figure S1) and this WWTP has only one inlet."

Comment 3:

Lines 99-100: specify the maximum holding time—what is meant by “within a day”? Does that mean within 24 hours? Or that they were processed the same day as they were collected? Or could it be that samples were collected in the morning of one day and then processed after “a day”, like at the latest by the end of the next day (so up to 36 hours or so after sample collection)? Since the samples were 24-hour composites, then it may have been close to 48 hours or even more until samples were concentrated and extracted (depending on what is meant by “a day”).

A. Thank you for your comment. We have modified the text and now we are more specific about the holding time.

"All samples were transported to the lab on cooling boxes and with ice packages. Samples from aircraft and airport (Måby and Arlanda terminal 5) were processed for concentration and RNA extraction in the same day (less than 24 h after sampling), while samples from the six inlets of the WWTP were processed within 24 h after sampling."

Comment 4:

Lines 104-105: “Two independent replicates”—does this mean field duplicates (i.e., duplicate 24-hour composite samples were collected independently, then concentrated and extracted independently) or were they laboratory duplicates (i.e., the 24-hour composite samples were split into equal volumes, which were concentrated and extracted independently). Or does it mean that the extracted nucleic acids were analyzed independently using RT-qPCR (i.e., instrument replicates)?

A. Thank you for your comment. We have improved the description about the replicates as follows:

*"Two independent biological replicates were concentrated and RNA extracted per sample."
"Two independent technical replicates were measured for each biological replicate and control sample (positive, negative, and cross-plate controls) using qPCR. Results were accepted if the standard deviation between the technical replicates was less than 0.5."*

Comment 5:

I understand that tap water was used as a negative control, but were other typical RT-qPCR negative controls used? Like process control blanks, extraction blanks, no template controls, etc.?

A. Thank you for your question. Nuclease-free water was used as control blank and tap water was used as concentration/extraction blank. This information is stated in line 117-118 (submitted version).

Comment 6:

Was the new concentration procedure evaluated against other established procedures to determine the efficiency of recovery, the accuracy, etc.?

A. The concentration using the Promega kit versus a previously developed method by our group was evaluated in our two previous papers (see the references below). The concentration procedure of using 40 mL versus 80 mL of the initial sample was assessed by comparing the detection limits in the samples. As it is stated in the results section (lines 238-240): “*Despite using 80 mL of wastewater instead of 40 mL to increase viral concentrations, the N-gene/PMMoV ratios in aircraft samples remained near the detection limit during weeks 4, 7, 9, and 18.*”

References where the concentration procedures were evaluated:

- Jafferli MH, Khatami K, Atasoy M, Birgersson M, Williams C, Cetecioglu Z. Benchmarking virus concentration methods for quantification of SARS-CoV-2 in raw wastewater. *Science of the Total Environment* 2021; 755.
- Perez-Zabaleta M, Archer A, Khatami K, et al. Long-term SARS-CoV-2 surveillance in the wastewater of Stockholm: What lessons can be learned from the Swedish perspective? *Science of the Total Environment* 2023; 858.

Comment 7:

Likewise, it's not very clear on lines 108-111, how the samples were processed. The authors state “four independent samples” so does that mean there were four autosamplers at each location (I'm guessing not), or that the composite sample volume was split into four 40 mL aliquots (i.e., laboratory processing duplicates)? Then, I assume that the concentrated volumes from two 40 mL aliquots were mixed (producing duplicate composite sample concentrates), which were each then extracted independently (producing duplicate nucleic acid extracts from four 40 mL aliquots of a single composite sample). This should be made very clear in the methods section so that the reader does not have to guess or assume what was actually done.

A. Thank you for your observations. The paragraph has been modified as follows: “*Due to the relatively low SARS-CoV-2 content in aircraft samples, a new concentration procedure was implemented from week 5 onwards to enhance the likelihood of positive detection and increasing the amount of virus before RT-qPCR analysis. Four independent wastewater samples, each with a volume of 40 mL, were concentrated and eluted to a final volume of 500 μ L using the Promega kit. The four resulting concentrates of 500 μ L each were then divided into two groups, and the samples within each group were combined, yielding approximately 1000 μ L per group. These combined samples were subsequently used for RNA extraction, producing two independent biological replicates, each with a volume of 80 μ L.*”

Comment 8:

It would have been more clear to specify the type of RT-qPCR used earlier in the manuscript (e.g., SYBR, probe-based, etc.). I figured out it was SYBR from reading below in the methods section about analysis of melt curves, but it should have been stated more clearly, and it should have been justified (why SYBR instead of probe-based RT-qPCR?). Also, the justification for using the N3 primers instead of or in addition to N1 and/or N2 did not appear to be very strong to me. Most other wastewater surveillance studies have used N1 and N2 in addition to or instead of N3, and many have reported preference for N1 and N2 (often analyzed in duplex). The supplementary materials document describes the approach used to compare the three primer sets targeting the N gene, and Figure S1 does indicate that N3 produced higher concentrations than N1, but it also indicates that N2 produced higher concentrations than N1, and there does not seem to be any significant difference between N2 and N3 concentrations. Therefore, I find

the justification for using N3 to be lacking (especially considering that it goes against the assays that have been used in numerous other studies).

A. Thank you for your comment. We have mentioned earlier in the manuscript that the method used was SYBRGreen. *“SARS-CoV-2 contents were determined via reverse transcriptase quantitative polymerase chain reaction (RT-qPCR) using SYBR Green one-step kit (Bio-Rad)...”*

We began monitoring SARS-CoV-2 in Stockholm’s wastewater at the very start of the pandemic (April 2020) using the SYBRGreen method. After several months of weekly surveillance and as the availability of reagents improved, we compared the detection capabilities of SYBRGreen versus TaqMan. During the tested period of around three months, we analysed samples from six inlets in Stockholm using both methods and found no clear differences between them.

Furthermore, to verify if the TaqMan method could enhance detection during aircraft monitoring, we tested aircraft samples with both TaqMan and SYBRGreen methods again. We observed no improvement or clear differences between the two methods. Initially, we did not include this comparison in the manuscript because the focus of our study was on the detection of variants rather than the comparison of methods or primers. However, in response to the reviewer’s query, we have now included a figure comparing the methods and statistical analysis of the differences in the supplementary material (Figure S4).

The following paragraph was added to the Method section: *“SARS-CoV-2 concentrations in aircraft samples were near the detection limit for several weeks during the monitoring period. For this reason, we evaluated whether the detection limit could be improved using TaqMan method or if there was any difference among the well-known primers targeting the N-gene (N1, N2 and N3 primers). The methodology and results are presented in supplementary material.”*

In addition, we incorporated a motivation why N3 was the preferred primer and SYBRGreen as detection method:

“The N3 primers were preferred over N2 due to evidence from another study indicating that N3 exhibited superior performance and better detection limits with stool samples.”

Furthermore, the detection of SARS-CoV-2 in aircraft samples was evaluated using both the TaqMan and SYBRGreen methods. The comparison revealed no clear differences between the two methods (Figure S4). At times, both methods detected similar amounts of SARS-CoV-2, while in other instances, TaqMan reported lower or higher levels than SYBRGreen. The SYBRGreen was preferred over TaqMan because with suitable primer sets such as N3, SYBRGreen has been shown to perform comparably or even better than the TaqMan method in terms of sensitivity, which is crucial for detecting low levels of viral RNA in wastewater samples.”

References:

Lu X, Wang L, Sakthivel SK, *et al.* US CDC real-time reverse transcription PCR panel for detection of severe acute respiratory syndrome Coronavirus 2. *Emerg Infect Dis* 2020; **26**. DOI:10.3201/eid2608.201246.

Tao Y, Yue Y, Qiu G, Ji Z, Spillman M, Gai Z, Chen Q, Bielecki M, Huber M, Trkola A, Wang Q, Cao J, Wang J. Comparison of analytical sensitivity and efficiency for SARS-CoV-2 primer sets by TaqMan-based and SYBR Green-based RT-qPCR. *Appl Microbiol Biotechnol*. 2022

Mar;106(5-6):2207-2218. doi: 10.1007/s00253-022-11822-4. Epub 2022 Feb 26. PMID: 35218386; PMCID: PMC8881549.

Comment 9:

The authors stated (lines 124-125) that reactions “were considered positive if the cycle threshold was less than 40 cycles with a single melting peak at the correct temperature.” This is not described correctly. Probably it was meant that the fluorescence crossed the established threshold before 40 cycles, but how was the threshold set? Was it set manually, or automatically (using software)? If software was used, which one? Was a sample considered positive if only one of two replicates amplified? How was the limit of detection determined? What about the limit of quantification? These should both be reported for all assays used. These are important details that must be considered and described in the methods section. Bustin et al. 2009 (DOI: 10.1373/clinchem.2008.112797) provide recommendations about the minimum information that should be provided when using qPCR or RT-qPCR, and this manuscript lacks some of this information.

A. Thank you for your comment. We have modified the sentence and added information about the threshold set as follows: “*Reactions were considered positive if the fluorescence crossed the established threshold before 40 cycles (if Ct was less than 40) and if a single melting peak was observed at the correct temperature. The threshold was set automatically using CFX Manager™ Software.*”

We have carefully gone through the required information and now the manuscript states all the information following the MIQE guidelines. The sections of the manuscript where the MIQE information was incorporated are indicated in parentheses:

1. **Experimental Conditions:** Describe the sample source and specify the experimental design (*Wastewater sampling*). Detail the RNA/DNA extraction method and quality assessment (*Wastewater concentration and RNA extraction and Weekly monitoring of SARS-CoV-2*).
2. **Assay Characteristics:** Provide primer and probe sequences (*Weekly monitoring of SARS-CoV-2*). Specify the qPCR instrument and cycling conditions (*Weekly monitoring of SARS-CoV-2*) and describe the reverse transcription (RT) step if applicable (*Weekly monitoring of SARS-CoV-2*).
3. **Reagents and Materials:** List all reagents used (*Wastewater concentration and RNA extraction and Weekly monitoring of SARS-CoV-2*), Mention the reference gene(s) used for normalization (*PMMoV gene, Weekly monitoring of SARS-CoV-2*).
4. **Data Analysis:** Explain the method for quantification (standard curve, *Weekly monitoring of SARS-CoV-2 and Calculations*)

Information about the LOD and LOQ was added to the main text in the manuscript:

“*The limit of detection (LOD) and limit of quantification (LOQ) of the qPCR assays used for N3 were 11.7 copies/μL and 35.4 copies/μL, respectively, and for PMMoV, they were 22.8 copies/μL and 69.0 copies/μL, respectively.*”

Thank you for your question regarding the criteria for considering a sample positive. A sample was deemed positive if at least three out of four measurements were positive, which includes two technical replicates for each of the two biological replicates. If only two out of four measurements were positive, the samples were re-run by qPCR. If the result remained the same upon re-testing and the melting curves did not show clear signals indicating positive/negative

results, the samples were concentrated again and tested once more by qPCR. This information has been added to the manuscript.

Comment 10:

For example, information about the standard curves is lacking (I see plasmids were used for PMMoV, but what about SARS-CoV-2 N3?). Also, what is meant by a “control sample of RNA from wastewater” that was used for reference and quality control? Presumably, this is RNA from a wastewater sample that tested positive for N3, but it’s unclear how it can be used as a control, especially if matrix spikes or other types of inhibition controls were not performed.

A. Standard curves were created using SARS-CoV-2 DNA (2019-nCoV_N_Positive Control, IDT, Cat. 10006625), and a constructed plasmid containing the appropriate target for PMMoV (IDT, custom MiniGene 25-500 bp). This is stated in the manuscript, line 118-121 (submitted version).

The control sample, named the cross-plate control, is a quality control sample with known concentrations of N3 and PMMoV. We used this sample to ensure that the observed values were comparable to the expected values, with no significant deviation. This control, along with the positive control, helps identify any malfunction in the qPCR analysis.

Comment 11:

Lines 116-117 and 127-131: There is not yet consensus in the literature for using PMMoV to normalize data. See Mazumder et al. 2022 (DOI: 10.1016/j.coesh.2022.100363) and Greenwald et al. 2021 (DOI: 10.1016/j.wroa.2021.100111) as two examples where multiple normalization factors were tested (including PMMoV, crAssphage, and others). Neither group found evidence that PMMoV normalization increased correlation with case numbers. In fact, one of them reported that crAssphage was more appropriate for normalization. These two studies are only examples. There are also many other studies that have examined normalization factors for wastewater surveillance.

A. We agree that there is not consensus in the literature. However, our previous publication (Perez-Zabaleta et al., 2023) provides a comprehensive analysis of SARS-CoV-2 data normalization using PMMoV and flow rates. This study, which spanned over one and a half years and included data from six inlets in Stockholm, demonstrated that PMMoV is an effective method for normalizing fluctuations caused by factors such as melting snow and stormwater. Based on these findings, we have applied the same normalization method in this paper.

Reference:

Perez-Zabaleta M, Archer A, Khatami K, *et al.* Long-term SARS-CoV-2 surveillance in the wastewater of Stockholm: What lessons can be learned from the Swedish perspective? *Science of the Total Environment* 2023; **858**. DOI:10.1016/j.scitotenv.2022.160023.

Comment 12:

For the calculations section (126-138), it is unclear how the “total SARS-CoV-2 detected in the Stockholm region” was calculated from various inlets. Were flow rates measured? Were loadings calculated? Were concentrations averaged? Unless you’re measuring flow rates and calculating loadings, you’re potentially getting a biased estimate of “total SARS-CoV-2” (especially considering the potential limitations of using PMMoV for normalization, as described in comment above). It also seems like the normalization methods used were inconsistent between sampling sites (lines 134-138), which presents problems when comparing between sites. In the analysis of the PMMoV results, the authors noted a greater variability in

the concentrations of PMMoV in the aircraft samples compared to the WWTPs, and this was attributed to the “fluctuating faeces-to-water ratio as vacuum toilets have minimal water usage.” However, could it also have been attributed to the smaller population of individuals using the aircraft toilets, relative to the population being served by the WWTPs? PMMoV is a pepper pathogen, so the presence of PMMoV in human feces is dietary based, and if someone does not eat peppers, it would be logical to assume that their feces would not contain PMMoV. Since the population on each aircraft is much much smaller than the populations served by the WWTPs, if only a fraction of the population sheds PMMoV, it would logically be more likely to see Poisson-like variations in the number of people shedding PMMoV in aircrafts, and thus higher variability in the concentrations of PMMoV in aircraft wastewater.

A. We appreciate the opportunity to clarify the methodology used in our study.

Calculation of “total SARS-CoV-2 detected in the Stockholm region”: The total SARS-CoV-2 was calculated using flow rates and PMMoV normalization as detailed in our previous publication (Perez-Zabaleta et al., 2023). The calculation details are provided in a very detailed way in our previous publication, and we did not include them again in this paper to avoid redundancy, as the main aim of this paper is the variant analysis.

Normalization Methods: The data is normalized with the flow rate and PMMoV. We have not normalized the data of the aircraft and airport region because this sampling site doesn’t have the flow rate data.

Figure S6: For reference, Figure S6 shows the data of the five locations without normalization (PMMoV or flow rate).

Regarding the variability in PMMoV concentrations in aircraft samples, you raise a valid point about the smaller population size potentially contributing to greater variability. While the fluctuating faeces-to-water ratio due to vacuum toilets with minimal water usage is a factor, the smaller population size on aircraft could indeed lead to Poisson-like variations in the number of people shedding PMMoV, resulting in higher variability in the concentrations of PMMoV in aircraft wastewater. Another study could investigate PMMoV normalization in smaller population sizes and with different types of samples, such as those from aircraft.

Additionally, we have added the following sentence to the manuscript to address the previously unmentioned Figure S6: *“For reference, Figure S6 in the supplementary material presents the data from the five locations without normalization (PMMoV or flow rate).”*

Comment 13:

My expertise in RNA metagenomic sequencing is more limited relative to my experience with the use of RT-qPCR, but I saw fewer issues with the description of the methods used for sequencing and data analysis, except that there did not appear to be any negative or positive controls.

A. Thank you for your observation. We have utilized an external service for sequencing provided by the Uppsala Genome Center (Science for Life Laboratory, Dept. of Immunology, Genetics and Pathology, Uppsala University). Since the emergence of new variants in 2021, we have been collaborating with the National Genomics Infrastructure (NGI) Sweden, particularly the laboratories in Uppsala. They have enhanced the sequencing process for wastewater samples, and we have contributed samples to aid in method development.

As the sequencing was conducted as an external service, we did not supply any additional positive or negative controls. NGI Sweden is an accredited clinical laboratory with both ISO and SWEDAC certification. More information can be found at:

<https://ngisweden.scilifelab.se/about-us/accreditation-certification/>

Comment 14:

All things considered, there are too many details lacking and limitations associated with the methods section to have confidence in the quality of the data. The interpretation of some of the results also lacked statistical rigor. As such, conclusions drawn are severely limited due to the lack of strong and scientifically sound evidence.

A. We understand your concerns regarding the methods section and the interpretation of the results. We would like to address these points as follows:

Most of the comments from the reviewer pertained to the qPCR analysis, a methodology that has been discussed in detail in our previous publications on SARS-CoV-2. These details were not included in this paper as the primary aim was to investigate and compare SARS-CoV-2 variant distribution at different sectoral levels, from aircraft to city level. The reviewer did not raise any relevant comments about the sequencing and variant study, which was the main focus of the paper.

However, we have now included more detailed information about the qPCR method in the manuscript to provide clarity and address the reviewer's concerns. We have made an effort to balance the amount of text to avoid redundancy while ensuring that the methodology is sufficiently detailed for the reader to have confidence in the quality of the data.

Comment 15:

Regarding the analytical approach:

The authors used Pearson correlation analysis (line 167), but did not describe if the data were normally distributed. If data are not distributed normally, then Spearman's correlation analysis may be more appropriate. This is a limitation. Also, on lines 206-207, the authors noted a lack of notable differences when comparing Käppala to the other WWTP inlets. Besides visually inspecting Figure S4, it does not appear that any statistical analysis was done to assess potential differences. The analytical approach is also severely limited by the lack of appropriate experimental controls, such as matrix spikes or RT-qPCR inhibition controls.

A. Thank you for your comment. Based on the Shapiro-Wilk test, the data for Stockholm and Käppala follow a normal distribution. Therefore, Pearson correlations are appropriated for this analysis.

We thank the reviewer for pointing out where we could make our arguments stronger when comparing Käppala to the other WWTP inlets. We have now run Procrustes tests between Käppala and each of the other inlets, which all showed symmetric correlations in the 0.95-0.97 range, with all p-values = 0.001 on 999 permutations. The following sentence was added to the manuscript: *"There was a very high correlation between Käppala and the other inlets of the WWTPs in Stockholm (all symmetric Procrustes correlations > 0.95, all p < 0.001)"*

As we have detailed in comments 1, 5, and 10, our analytical approach does account appropriate experimental controls. We addressed the use of RT-qPCR negative controls (nuclease-free water), RNA extraction negative control (tap water) to ensure the accuracy of our quantification. We elaborated on the steps taken to validate our method, including the use of appropriate experimental controls such as PMMoV positive controls, SARS-CoV-2 commercial positive control and cross-plate controls.

These points collectively demonstrate that our approach is robust and includes the necessary controls to validate our findings. We hope this clarifies any concerns regarding the experimental design and the measures taken to ensure the reliability of our results.

Comment 16:

Conclusions:

In summary, the shortcomings described above severely limit the potential significance of the findings presented herein. This is especially the case, considering how many other studies have drawn similar conclusions over the past four years. For example, the conclusion that wastewater surveillance can detect “more variant diversity” and that it was “more effective ... in the early detection of specific variants” is not a novel finding, and the authors did not cite previous studies that have drawn similar conclusions. There are many studies that have drawn these conclusions, but here are just a few from 2022 as examples:

Vo et al. (2022). Use of wastewater surveillance for early detection of Alpha and Epsilon SARS-CoV-2 variants of concern and estimation of overall COVID-19 infection burden. *Science of The Total Environment*, 835, 155410.

Jahn et al. (2022). Early detection and surveillance of SARS-CoV-2 genomic variants in wastewater using COJAC. *Nature Microbiology*, 7(8), 1151-1160. **CITED**

Karthikeyan et al. (2022). Wastewater sequencing reveals early cryptic SARS-CoV-2 variant transmission. *Nature*, 609(7925), 101-108. **CITED**

A. We respectfully disagree with the reviewer’s comment, as two of the three papers were already cited in the submitted manuscript (see lines 50-52 in the previous version) and we have mentioned that WBE can detect SARS-CoV-2 variants earlier than clinical surveillance. We have now included the third reference (Vo et al.) in the revised manuscript.

Comment 17:

The inclusion of aircraft wastewater is a component of this manuscript that has been less studied in previous works, but there have been other aircraft wastewater surveillance studies, and the discussion section lacks a comparison of this study’s results to other studies of aircraft wastewater surveillance. Here are a few examples of these studies.

Ahmed et al. (2020). Detection of SARS-CoV-2 RNA in commercial passenger aircraft and cruise ship wastewater: a surveillance tool for assessing the presence of COVID-19 infected travellers. *Journal of Travel Medicine*, 27(5), 1-11.

Morfino et al. (2023). Notes from the field: aircraft wastewater surveillance for early detection of SARS-CoV-2 variants—John F. Kennedy International Airport, New York City, August–September 2022. *MMWR. Morbidity and Mortality Weekly Report*, 72. **CITED**

Jones et al. (2023). Suitability of aircraft wastewater for pathogen detection and public health surveillance. *Science of The Total Environment*, 856, 159162. **CITED**

A. Two of the three papers were already cited in the submitted manuscript. We have now included in the revised manuscript a comparison of the other studies on aircraft with our current study to highlight the novelty of our research.

Comment 18:

All things considered, the results from this study do not significantly advance understanding about the science of wastewater surveillance and lack the potential to move the field forward significantly.

A. We would like to emphasize that the main focus of our manuscript is the sequencing analysis and the discussion of SARS-CoV-2 variants, rather than the qPCR analysis. The sequencing analysis provides a comprehensive overview of the variant distribution, which is a significant contribution to the field of wastewater surveillance.

We believe that a more in-depth review of the SARS-CoV-2 sequencing data is crucial, as it offers insights into the prevalence and spread of different variants. This information is valuable for public health monitoring and can inform strategies for controlling the spread of the virus.

It's also worth noting that the reviewer did not raise any concerns about the sequencing analysis and discussion, which suggests that this aspect of the manuscript is sound and contributes to the understanding of SARS-CoV-2 dynamics in wastewater.

While the qPCR analysis is an important part of the methodology, we have provided additional details in the revised manuscript to address the reviewer's comments. However, we maintain that the sequencing analysis is the core of our study and offers significant insights that can advance the field of wastewater surveillance. We hope this clarifies the focus of our manuscript and the contributions it makes to the scientific community.

General Response

Thank you for the opportunity to revise and resubmit our manuscript. We appreciate the reviewers' thorough evaluation and constructive feedback. We are pleased to hear that the reviewers found our revisions to have improved the manuscript.

We have meticulously addressed the remaining concerns raised by the reviewers and have further refined the description of the qPCR methodology. Specifically, we have provided a detailed explanation to Reviewer 4 regarding the rationale behind the selection of the current qPCR methodology (SYBRGreen vs. TaqMan and N1, N2 and N3 primers). Additionally, we clarified that no controls were omitted; rather, they were referred to differently. While PCR inhibition controls are not required for each run, PCR inhibition testing is mandatory ¹. Consequently, we have included in the supplementary material an example of an inhibition test conducted on wastewater samples from Stockholm. Furthermore, we have enhanced the discussion and conclusion sections and revised the title to better emphasize the primary objective of our research.

1. Bustin, S. A. *et al.* The MIQE guidelines: Minimum information for publication of quantitative real-time PCR experiments. *Clin Chem* **55**, (2009).

Point-by-point responses to the comments.

REVIEWER COMMENTS

Reviewer #1 (Remarks to the Author):

General Comments:

The authors have addressed many of the issues raised in the previous review and the revised manuscript is much improved. I have noted a couple minor points below.

General Response

Thank you for your positive feedback on the manuscript. Below, we address to your observations point by point.

Specific Comments:

L148 - 149: Thank you for clarifying how replicates were analyzed. Please specify what quantity this standard deviation was calculated for. Was it the standard deviation of the Ct value between replicates? log₁₀ copies/uL? etc.

A Thank you for your comment. We have clarified this: “*Results were accepted if the standard deviation of the quantification cycle (Cq) values between technical replicates was less than 0.5.*”

L154 - 157: Thank you for including this LOD and LOQ information. I assume these values were determined in the authors' previous work. Because the definitions of LOD and LOQ vary widely between studies, please either define them here and describe how they were determined for these assays or explicitly direct the reader to the previous paper(s) where that information is provided. Additionally, it is not clear whether these limits are in terms of uL of reaction volume or uL or RNA template volume. Since a new concentration approach was implemented partway through the study with the explicit goal of improving detection sensitivity, the authors might consider converting these limits into copies/mL wastewater (or similar) for the two concentration approaches to provide a sense of the expected sensitivity improvements from the new concentration method.

A Thank you for your comment. We acknowledge that multiple methods exist for calculating the limit of detection (LOD) and limit of quantification (LOQ). We have revised the text to clarify these calculations and presented the LOD and LOQ as copies per PCR reaction, in accordance with the MIQE guidelines for qPCR. Additionally, we have included information on efficiencies, slopes, and R² values. We have converted LOD and LOQ into copy/ml wastewater as suggested by the reviewer. The following text has been added to the manuscript: “*The standard curves yielded calculated efficiencies of 98% and 90% for the N3 and PMMoV qPCR assays, respectively, with corresponding slopes of -3.38 and -3.59. The coefficient of determination (R²) values were 0.999 and 0.998, respectively. The limit of detection (LOD) and limit of quantification (LOQ) for the qPCR assays was determined using the standard deviation method, which involves the standard deviation of the response (y-intercepts of the regression lines) and the slope of the calibration curve. The LOD and LOQ of the qPCR assays used for N3 were 0.4 copies/ml of wastewater sample and 5.3 copies/ml, respectively, and for PMMoV, they were 0.4 copies/ml and 4.1 copies/ml, respectively.*”

Reviewer #2 (Remarks to the Author):

I greatly appreciate authors' intensive efforts to revise this manuscript according to reviewers' comments. I think the presentation of the manuscript has been significantly improved, but some minor changes are still required. One of the conclusions reached by authors is that wastewater monitoring is more effective than clinical testing in early detection of specific variants, with notable delays observed in the clinical surveillance (L28-29). However, when targeting specific variants, like Variants Under Monitoring (VUMs), Variants of Concern (VOCs), and Variants of Interest (VOIs) in wastewater, RT-qPCR rather than next-generation sequencing is sufficient for viral sequence detection. This point should be more explicitly described in the text, including abstract. Regarding the required number of wastewater treatment plants (WWTPs) to perform city-wide infection assessment, this is highly dependent on local conditions, including the size of the sewer-shed (population coverage) and the type of sewer system (separated or combined). In some cases, a single large WWTP covering 70-80% of the population in a large city may be sufficient for monitoring. More detailed discussion of this issue is necessary.

General Response

Thank you for your insightful comments on our study. We have revised the abstract to explicitly highlight that RT-qPCR is sufficient for detecting specific variants in wastewater, such as Variants Under Monitoring (VUMs). However, we also acknowledge that next-generation sequencing is a potent tool for detecting new variants. This revision emphasizes the efficiency and practicality of RT-qPCR, while also recognizing the valuable role of next-generation sequencing in wastewater surveillance.

Regarding the WWTPs, we have added a paragraph to the discussion section to include this information. Additionally, we have incorporated the following details into the materials and methods section:

“Stockholm has an urban population of around 1.6 million and a metropolitan population of approximately 2.4 million. Samples were collected from three main municipal wastewater treatment plants in Stockholm: Bromma WWTP (serving around 377,500 residents, 18% of the population), Henriksdal WWTP (serving about 862,100 residents, 41% of the population), and Käppala WWTP (serving around 700,000 residents, 33% of the population).”

Discussion:

“Our results on SARS-CoV-2 concentrations showed that wastewater surveillance from the WWTP (Käppala, which covers 33% of the population) only represents infections in that area and cannot be extrapolated to Stockholm city. For a complete picture of Stockholm’s infection, it is necessary to consider Bromma and Henriksdal WWTPs contents as well. However, when the SARS-CoV-2 variants were examined, the lineage profile detected in Käppala was similar to that of Stockholm. The number of WWTPs required for effective city-wide infection assessment is highly dependent on local conditions. Key factors include the size of the sewer-shed, which determines population coverage, and the type of sewer system, whether separated or combined. In some instances, a single large WWTP that covers 70-80% of the population in a large city may be sufficient for comprehensive monitoring.”

Reviewer #3 (Remarks to the Author):

Reviewer #4 (Remarks to the Author):

The revised manuscript is an improvement over the original submission. However, there are still some shortcomings. Below are my comments.

Both original reviewers were originally confused about the methodology, and several limitations were pointed out in the original review. Most of the confusion has been cleared up in the revised version of the manuscript, but there are still a few items that could use more clarification and there is one major limitation due to an omitted control.

The authors added more details to this part of the methods section and clarified that the equivalent sample volumes indeed increased due to concentration using a composited sample based on four independent replicates. On line 115, it should be stated that the manufacturer’s recommendations were followed, with some exceptions, because on line 124, the “new concentration procedure” (which is actually just a four-sample composite) appears to be a modification from the manufacturer’s recommended protocol. This is also the only reason why

the ESV is different (because instead of processing 500 μL as recommended in the protocol, a composited 1000 μL was processed). Also, instead of describing it as a “new concentration procedure”, I think it’d make more sense to call it a spatial composite of aircraft wastewater (which is probably highly heterogeneous in nature). So, by collecting 4 replicates and mixing them together, it’s essentially a composite sample. The authors should describe how the 4 replicates were selected (Did they all come from the same location in the wastewater tank? Or were some from the bottom of the tank and others from the top, etc.?).

A. Thank you for your comment. We have incorporated the reviewer’s suggestions to the manuscript on line 121-122 and line 131-132. The following sentence was also added to the manuscript: *“Aircraft wastewater samples were collected after thorough mixing during transport from the aircraft to the lavatory service truck. Samples were then taken from the top of the lavatory service truck immediately after the transfer.”*

It’s still not clear why N3 was selected over N2 or N1 and why SYBR green was selected over TaqMan. The authors stated on lines 258-259 that the comparison between SYBR green and TaqMan methods “revealed no clear differences between the two methods,” however after reviewing the supplementary information, there were huge differences between the CT values obtained using SYBR green vs. TaqMan, with some samples showing more than 5 CT units difference! This would equate to more than an order of magnitude difference in the concentrations. Based on those results, there is no valid justification for using SYBR green over TaqMan, especially given that TaqMan is a more specific method than SYBR green. Regarding the selection of N3 over the other targets, the authors cited one paper that reported this to be a better target but has ignored much of the literature that has used N1 and/or N2 over N3 (which constitutes the majority of studies).

A. Thank you for your comment. Since the beginning of the pandemic, we have been monitoring SARS-CoV-2 and conducting numerous tests to identify the most effective detection methods. When the N1 assay gained popularity for monitoring SARS-CoV-2 in 2021, we considered whether switching from our N3 SYBRGreen method to the N1 TaqMan method would be beneficial.

Our team was the only organization monitoring Stockholm, the capital of Sweden, from 2020 to 2023. We publicly reported our findings every Friday through a COVID portal in Sweden (https://www.pathogens.se/dashboards/covid_quantification/covid_quant_kth/) and on social media. Stockholm served as a reference point for the waves of infection, as most waves originated there before spreading to other cities in Sweden. Therefore, before making any changes to our methodology, we had to ensure that the N1 TaqMan method, as suggested by other researchers, was indeed superior. To achieve this, we simultaneously monitored Stockholm using both N1+TaqMan and N3+SYBRGreen for several weeks and conducted correlation studies with clinical cases (Figure A and B). Our study concluded that the N3+SYBRGreen method (Figure A) showed a higher positive correlation with clinical cases compared to the N1+TaqMan method (Figure B), which typically detected the peak of the wave one week later than clinical cases (case number) as it is presented in the following figures:

Figure A: Stockholm monitoring using SYBRGreen method and N3 primers. Clinical cases peak on week 3, 2022 and wastewater SARS-CoV-2 content peak on week 3, 2022

Figure B: Stockholm monitoring using TaqMan and N1 primers. Clinical cases peak on week 3, 2022 and wastewater SARS-CoV-2 content peak on week 4, 2022

This delayed peak detection was observed not only in Stockholm but also in Uppsala (Figure C), where N1 and TaqMan had been used since August 2020. In Uppsala, this method detected the peak of the wave two weeks later than clinical cases (case number).

Figure C: Uppsala monitoring using TaqMan and N1 primers. Clinical cases peak on week 4, 2022 and wastewater SARS-CoV-2 content peak on week 6, 2022

Given the lower correlation between clinical cases and wastewater SARS-CoV-2 content with the N1+TaqMan method, we decided to continue using the N3 SYBRGreen method. Although these results were not published, they represent extensive work conducted during the pandemic, which we were unable to publish in full due to time constrain. However, all the decisions reflected in our current manuscript is based on the experience gained since we began monitoring SARS-CoV-2 in wastewater in April 2020.

In the current manuscript, we have also tested different primers (N1, N2 and N3) and methods (SYBRGreen and TaqMan) to reconfirm that using N3 and SYBRGreen is optimal for aircraft samples, which are unique due to their low water content and characteristics.

Figure S3 in the supplementary material compares different primers (not detection methods). It clearly shows that N1 detects lower amounts of SARS-CoV-2 than N3 and N2, with more pronounced differences in Kappala than in aircraft samples. Concluding that N3 and N2 are superior to N1.

Figure S4 evaluates the detection methods, SYBRGreen vs. TaqMan. Although some samples exhibit more than a 5 Cq unit difference as the reviewer stated, the variability is considerable and has to be taken into account. There is no consistent pattern in the results; sometimes the Cq values for N1 are higher, and sometimes they are lower. Consequently, we cannot definitively determine which method is superior. Statistical analyses indicate no significant difference between the detection methods, leading us to state in the manuscript: “revealed no clear differences between the two methods.” Therefore, combined with our results from early 2022, we concluded that maintaining our qPCR analyses (N3 SYBRGreen) is the best decision.

The results of the standard curves are not disclosed. Things like the slope, intercept, R^2 value, and most importantly, the efficiency (which should be between 90% and 110%). This is important to report especially since the SARS-CoV-2 assays often had poor efficiencies in other published studies (sometimes they were much greater than 110%).

A. Thank you for the feedback. We have included the required information in the manuscript (line 171-179):

“The standard curves yielded calculated efficiencies of 98% and 90% for the N3 and PMMoV qPCR assays, respectively, with corresponding slopes of -3.38 and -3.59 . The coefficient of determination (R^2) values were 0.999 and 0.998, respectively. The limit of detection (LOD) and limit of quantification (LOQ) for the qPCR assays was determined using the standard deviation method, which involves the standard deviation of the response (y-intercepts of the regression lines) and the slope of the calibration curve. The LOD and LOQ of the qPCR assays used for N3 were 0.4 copies/ml of wastewater sample and 5.3 copies/ml, respectively, and for PMMoV, they were 0.4 copies/ml and 4.1 copies/ml, respectively.”

In general, part of the issue with the confusion about the methods and the lack of confidence in the results is that the authors do not follow the widely-cited MIQE guidelines for the publication of results from qPCR (Bustin et al. 2009; doi: 10.1373/clinchem.2008.112797). This minimum information should be disclosed in all publications that use qPCR, according to that study, which has thousands of citations.

A. We agree that the MIQE guidelines should always be followed. We recognize that the initial version of the manuscript omitted several details about these guidelines because we aimed to emphasize variant detection over qPCR analyses. This decision was also influenced by our previous publication, which focused solely on qPCR analysis in Stockholm, and we wanted to avoid being repetitive in the description of the methods. However, after receiving constructive feedback from the reviewers, we realized the importance of including all relevant details, even if it may seem redundant. Consequently, we have made a concerted effort to incorporate the required information. We have thoroughly reviewed the MIQE checklist to ensure that all necessary details are included in our manuscript.

In the revised methods section and supporting information document, the authors added many details about many of the normal controls used during qPCR (nuclease free water in qPCRs, which is commonly called a “no template control”, as well as details about the plasmids used as the positive control to construct the standard curve), however many other typical qPCR controls are missing or were done in a non-conventional way. For example, the tap water used as a negative control is not a conventional way to run a negative control. The cross-plate control is a nice QA/QC addition, but it is unclear what the authors did with the information from these controls. Presumably the CT values were not vastly different from plate to plate, but what was the range of differences observed in the cross-plate controls? That should be reported, at least in the Supporting Information document. Many typical controls were omitted. Typically, field blanks, process blanks, and extraction blanks, consisting of reagent water, MilliQ water, or nuclease free water, are analyzed alongside samples to determine if there was contamination in the field, during sample concentration, or during nucleic acid extraction. The authors stated that the tap water negative controls “performed as expected” and presumably that means there was no amplification at all, but this was not explicitly stated.

A. Thank you for the comment. We did not omit typical controls and have included all recommended controls according to MIQE guidelines. Tap water is a negative control, which

serves as process/extraction control. As the reviewer noted, controls can consist of reagent water, MilliQ water, or nuclease-free water. We have chosen tap water as our reagent water. We have enhanced the description of each control to improve clarity for readers lines 150-152 and 157-163.

“Nuclease-free water was included as no-template control (NTC) for all qPCR reactions.”

“Two tap water samples were concentrated, and RNA was extracted and analysed alongside the weekly samples to assess potential contamination during handling.”

“A control sample of RNA from wastewater (cross-plate controls) with known SARS-CoV-2 and PMMoV concentrations was used in each qPCR analysis for reproducibility and quality control. Cross-plate controls were stored at -80°C in aliquots. A new batch was prepared when the quantification cycle (C_q) shift of 0.5-1 was detected, with each batch typically used for up to eight weeks. The cross-plate control was employed to evaluate the long-term precision, which refers to the variation in results between runs, in accordance with MIQE guidelines¹. The standard deviation of the cross-plate controls was calculated between runs. Results were accepted if the standard deviation of the C_q values was less than 0.5.”

In addition, we have modified the sentence: “The PCR negative and positive controls performed as expected, indicating the reliability of the procedure.” and included more information (lines 252-255):

“The qPCR negative controls (tap water and NTC) showed no amplification, confirming the absence of contamination. The positive controls (SARS-CoV-2 IDT, PMMoV IDT and cross-plate) produced the expected amplification results, verifying the accuracy and efficiency of the qPCR procedure. These results demonstrate the reliability of the qPCR process.”

The most important control that is missing in this study is the PCR inhibition control. Wastewater has numerous potential PCR inhibitors which can change from day to day and sample to sample, potentially introducing major biases into the interpreted results due to PCR inhibition (which would make the concentration appear to be lower than it actually might have been). PCR inhibition is most commonly assessed using the TaqMan method with an internal amplification control (IAC), which is an amplicon with the same primer pair but a different probe sequence that a known quantity of which is added to a qPCR replicate for every sample. Or, alternatively, an additional replicate of every sample is analyzed with a known quantity of standard (e.g., plasmids) added to the qPCR reaction well (or some people analyze an additional replicate but with a dilution of the template RNA). If the C_q value does not change as expected with the plasmid spike or the diluted template RNA, then it is a sign of inhibition. Without some type of PCR inhibition controls, the reliability of the data is limited.

A. Thank you for the comment. There are various methods to evaluate PCR inhibition. The reviewer mentioned that "PCR inhibition is most commonly assessed using the TaqMan method with an internal amplification control (IAC)," which is just one approach. We have assessed inhibition by diluting wastewater samples as recommended by MIQE guidelines and calculating the calibration curve and PCR reaction efficiency.

Extracted from MIQE guidelines:

The potential for inhibition is a more generally applicable variable that must be addressed in a publication, and it is important to ensure that no inhibitors copurified with the DNA will distort results, e.g., pathogen detection and their quantification (55). Although such approaches such as spiking samples with positive controls (52) can be used to detect inhibition, different PCR reactions may not be equally susceptible to inhibition by substances copurified in nucleic acid extracts (56, 57). Consequently, it is better to routinely use dilutions of nucleic acids to demonstrate that observed decreases in C_q s or copy numbers are consistent with the anticipated result and to report these data.

We agree that evaluating PCR inhibition in wastewater samples is crucial. Therefore, our team have conducted inhibition tests since the beginning of our monitoring. For instance, inhibition tests were included in the main manuscript of Jafferli et. al 2020 (<https://doi.org/10.1016/j.scitotenv.2020.142939>). In subsequent papers, we have only mentioned that inhibition tests were performed. However, this time, we recognize that more information about the inhibition test must be provided. We have conducted inhibition testing on wastewater samples from various inlets and pooled samples from the Stockholm region. Therefore, we have included an example of the inhibition testing in the supplementary material and provided a more detailed explanation in the materials and methods section.

Please note, according to the MIQE guidelines, inhibition controls are not required for each run; however, routine inhibition testing is necessary (Table 1. MIQE checklist for authors, reviewers, and editors ¹). The following sentence was included in the manuscript (line 179-182):

“Inhibition testing was conducted utilizing the C_q dilution method, as recommended by the MIQE guidelines. The results demonstrated high efficiencies and strong R^2 correlation coefficients for the standard curve. An example of the inhibition test is provided in the supplementary material (Figure S9).”

Figure included in the supplementary material:

Figure S9: Inhibition testing on a pooled sample from Stockholm. The samples were subjected to a 10-fold serial dilution. Calibration parameters, including R^2 correlation, slope, and qPCR efficiency, were calculated to assess the accuracy and reliability of the inhibition testing

DISCUSSION:

The revised manuscript had minimal revisions to the discussion section. A previous reviewer brought up the observation that the authors did not discuss their results within the context of the findings from previously published aircraft wastewater surveillance studies. These previous studies are cited (in the introduction), but there is no discussion about how the results from the present study advance the knowledge of what has been reported previously, including in studies of aircraft wastewater. The revised manuscript still states (lines 307-309) that “the influence of SARS-CoV-2 contents and variants of aircraft and airport regions on urban areas remains unexplored.” Ahmed et al. (2020) detected the virus in aircraft wastewater five years ago. How did this study advance their work? Morfino et al. (2023) used aircraft wastewater surveillance to identify SARS-CoV-2 variants. What was different about the findings of this study compared to that one? Were the conclusions similar? Different? Jones et al. (2023) did a thorough assessment of the utility of aircraft wastewater surveillance, quantifying toilet use/behavior on flights of different durations. How were the findings from that study incorporated into the present study? Did the authors consider the duration of the flights? Why or why not, given the findings from Jones et al. (2023). I’m not suggesting that the authors respond to each of these questions posed here, these are just examples of how one might discuss the results of a study in comparison/contrast to previously published papers on the topic. My point is that the discussion section of this manuscript is limited in that it does not relate the findings to any of the previously published findings from studies of aircraft/airport wastewater surveillance.

A. Our main goal with the discussion is not to analyse the intricacies of analysing aircraft wastewater. This has been done previously, and we do indeed cite the works mentioned above. Our work, however, focuses on comparing (one example of) aircraft-based WBE with WWTP-based WBE and clinics-based epidemiological surveillance. To make this clearer, we have opted to remove one paragraph of the discussion that placed exaggerated emphasis on aircraft data only. Additionally, we have added the following sentences for clarification:

” This also suggests that, for monitoring, a balance must be achieved between coverage and granularity of the wastewater sources being analysed. Single-person and single-aircraft approaches do not scale well, while we also found that combining all samples before analysis was not as sensitive as analysing each inlet separately.” (lines 385-388)

“However, even in long-haul flights such as the ones analysed here, not all travellers defecate during the flight, which could influence the representativeness of the sample [Jones et al 2023; Morfino et al 2023]. Therefore, broadening the scope of the sampling location could improve sensitivity and decrease the time to first detection, as defined by St-Onge *et al* [St Onge et al 2025].”

CONCLUSIONS:

The revised manuscript had no revisions to the conclusions section. A previous reviewer reported concerns about the lack of novelty in the conclusions. Papers published by Nature Communications journal are typically very high impact studies that represent “important advances of significance” to specialists within a particular field. The conclusions of the revised manuscript fall short of this expectation. The data from the metagenomics analysis indeed

showed that wastewater surveillance can detect “more variant diversity” earlier than clinical surveillance. It is important to study variants of concern and to have a way to make early predictions about their dissemination. I don’t disagree that this study showed that wastewater surveillance has the potential to provide this important information earlier than clinical surveillance, and this would have been a very impactful and novel finding in 2020 or 2021. However, at this point (now more than 5 years after the start of the pandemic), the conclusion is only confirming the conclusions from a plethora of previous studies, many of which were published years ago. The authors indeed cited some of these previous studies, but adding more citations to the manuscript does not change this fact that the conclusions of this study do not represent an “important new advance of significance” to the field.

A. Thank you for your comment. One novel aspect that our study addresses and puts into application is the possibility to apply wastewater surveillance in a variety of settings (and during the same time period) at different levels that otherwise would require more efforts to cover with traditional methods such as collecting and analysing a large number of individual clinical samples. Especially in terms of scalability, we demonstrate that the same methodology can be applied in the mentioned settings delivering valuable information in a timely manner. We emphasized this in the revised version by pointing out the large potential that lies in the scalability and flexibility of wastewater surveillance also underlining the yet under-explored possibility of using one sample for many different analyses at the same time. We also slightly modified the title of the study to further underline those aspects.